# WORLD MODEL ON MILLION-LENGTH VIDEO AND LANGUAGE WITH BLOCKWISE RINGATTENTION

**Hao Liu**[*] **Wilson Yan**[*] **Matei Zaharia**   **Pieter Abbeel**

UC Berkeley

## ABSTRACT

Enabling long-context understanding remains a key challenge in scaling existing sequence models – a crucial component in developing generally intelligent models that can process and operate over long temporal horizons that potentially consist of millions of tokens. In this paper, we aim to address these challenges by providing a comprehensive exploration of the full development process for producing 1M context language models and video-language models, setting new benchmarks in language retrieval and new capabilities in long video understanding. We detail our long context data curation process, progressive context extension from 4K to 1M tokens, and present an efficient open-source implementation for scalable training on long sequences. Additionally, we open-source a family of 7B parameter models capable of processing long text documents and videos exceeding 1M tokens.

## 1 INTRODUCTION

Enabling long-context understanding remains a key challenge in scaling existing sequence models—a crucial step toward developing generally intelligent models that can process and operate over extended temporal horizons, potentially involving millions of tokens. Current modeling approaches are predominantly limited to processing short sequences, whether in the form of language, images, or video clips (Brown et al., 2020; Touvron et al., 2023a;b; OpenAI, 2023; Brooks et al., 2024; Team et al., 2023). As a result, these models fall short when tasked with understanding complex, long-form language and visual contexts.

However, training models to process sequences that exceed millions of tokens is a significant challenge due to the high memory and computational costs, as well as the lack of long-context data. In this work, we address these challenges by leveraging Blockwise RingAttention (Liu et al., 2024; Liu and Abbeel, 2023), a technique that scales context size without approximations or overheads, enabling efficient training on long sequences. We curate an extensive dataset of long-form videos and books from

---

[*]Equal contribution. Email: `hao.liu@cs.berkeley.edu`, `wilson1.yan@berkeley.edu`
Code and models of Large World Model (LWM) are available at https://largeworldmodel.github.io/lwm/.

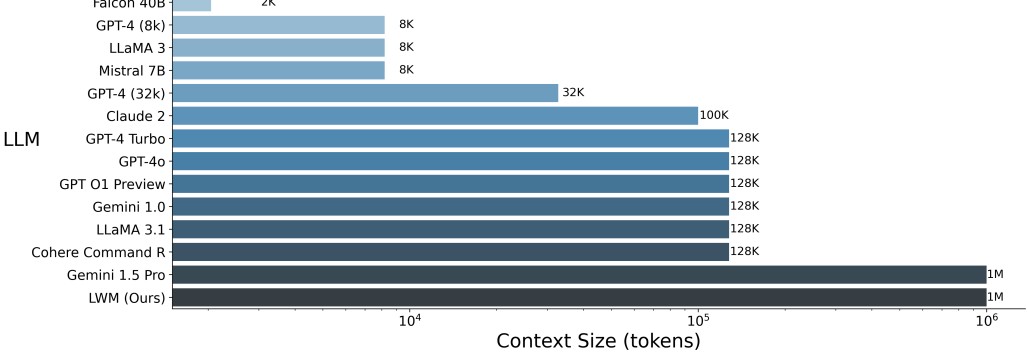

**Figure 1    Comparison of context size in state-of-the-art LLMs.** Our model and concurrent work Gemini 1.5 both achieve a 1M context size, significantly outperforming other LLMs.

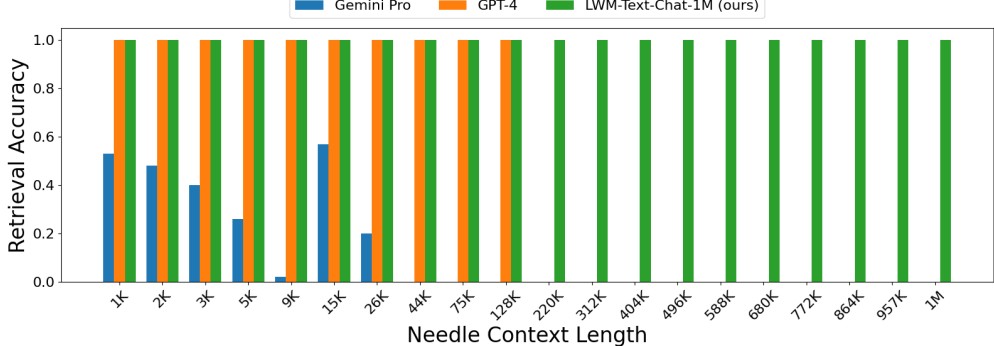

**Figure 2** **Retrieval comparisons against Gemini Pro and GPT-4.** Needle retrieval comparisons against Gemini Pro and GPT-4 for each respective max context length – 32K and 128K. Our model performs competitively while being able to extend to 8x longer context length. Note that in order to show fine-grained results, the x-axis is log-scale from 0-128K, and linear-scale from 128K-1M.

public sources, covering a wide variety of activities and narrative structures. To address the scarcity of long-form conversational datasets, we developed a model-based question-answering technique, where a short-context model generates training data from books, significantly enhancing the model's chat capabilities over long sequences. To mitigate computational costs, we gradually extended context size from an initial 4K tokens to 1M tokens, achieving a cost-effective and scalable approach for long-context modeling.

Following this, we further train our long-context language model to incorporate visual modalities, such as image and video. Contrary to existing popular vision-language models (Liu et al., 2023a; OpenAI, 2023; Chen et al., 2023a), we opt to additionally optimize next-token prediction losses for image and video (generation) with a VQGAN (Esser et al., 2021) encoder. We encountered various challenges training on mixed modalities (video, image, text). To balance their unique characteristics - sequential information, visual detail, and linguistic content - we implement an efficient masked sequence packing strategy, as well as introduce careful loss balancing to retain short context accuracy. This approach handles varying sequence lengths more effectively than standard methods. We also optimized the ratio of image, video, and text inputs in each batch, proposing an empirically effective balance for cross-modality learning. Since our model aims to model both textual and visual projections of the world through a large context window, drawing inspiration from prior work on world models (Brooks et al., 2024; Ha and Schmidhuber, 2018), we name our work as Large World Model (LWM).

Our contributions are threefold: (a) we train one of the largest context size transformers to date on long text documents and videos and achieved competitive results on long video understanding and long context fact retrieval. (b) We discover a range of challenges associated with training on long sequences and propose solutions for them: masked sequence packing to effectively train with different sequence lengths and synthetic model-generating question-answering for effective attention. (c) We provide an open-source and optimized implementation for training with millions of tokens in context, as well as a family of Llama-based 1M context models capable of processing long documents (`LWM-Text`, `LWM-Text-Chat`) and videos (`LWM`, `LWM-Chat`) of 1M tokens.

## 2 METHOD OVERVIEW

We train a large autoregressive transformer model with a large context window of up to one million tokens, building upon Llama2 7B (Touvron et al., 2023b). To achieve this goal, we implement a two-stage training strategy. In Stage I (Section 3), we extend the context to 1M tokens using book-length texts. This is followed by Stage II (Section 4), where we conduct joint training on diverse long multimodal sequences, incorporating text-image data, text-video data, and book-length texts. Our model architecture is the standard autoregressive transformer design, as illustrated in Figure 3. For a comprehensive overview of our training stages and the datasets employed, please refer to Figure 4.

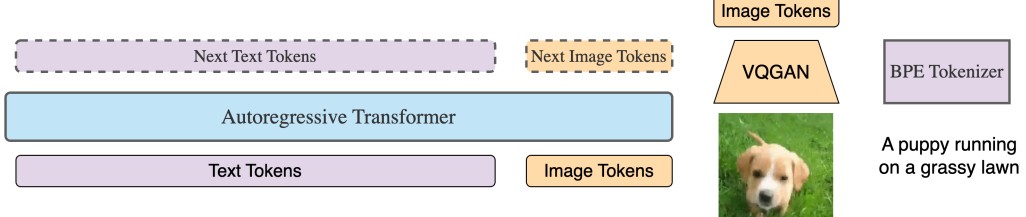

**Figure 3  Model Architecture.** The LWM model is an autoregressive transformer trained on sequences of multimodal tokens. Each video frame is tokenized into 256 tokens using VQGAN, while text is processed using a Byte-Pair Encoding (BPE) tokenizer. These tokens—both image and text—are combined and input into the transformer to autoregressively predict the next token. The model can handle various input-output modalities, including text, image, video, and text-video pairs. To distinguish between images and text, special tokens `<vision>` and `</vision>` are used for image and video frames, with `<eof>` and `<eov>` marking the end of these sequences. For simplicity, delimiters are not shown in the figure.

## 3  STAGE I: LEARNING LONG-CONTEXT LANGUAGE MODELS

This stage aims at first developing `LWM-Text` and `LWM-Text-Chat`, a set of long-context language models learned by training on progressively increasing sequence length data, and modifying positional encoding parameters to account for longer sequence lengths (see Section 3.1). In Section 3.2, we show how to construct model-generated question-answering data for enabling long sequence conversations.

### 3.1  PROGRESSIVE TRAINING TOWARDS LONG CONTEXT

Learning long-range dependencies over sequences of millions of tokens requires (1) memory efficient training to scale to such long sequences, as well as a need to (2) compute efficient training to extend the context of our base language model. We outline our approach to these challenges, detailing our methods for training on long sequences, designs for efficiency and stability, and experimental setup.

Training on long sequences has become prohibitively expensive due to memory constraints imposed by the quadratic complexity of attention weight computations. To address these computational limitations, we leverage recent advancements in scaling context window size, particularly Blockwise RingAttention (Liu et al., 2024). This approach theoretically allows for an infinite context, bounded only by available devices. We further enhance performance by fusing it with FlashAttention (Dao et al., 2022) using Pallas (Bradbury et al., 2018) to optimize performance compared with using XLA compiler. Notably, with enough tokens per device—already a given—the communication cost during sequence parallelism is fully overlapped by computation, resulting in no additional overhead.

For better efficiency, we adopt a training approach inspired by prior research on extending context (Jin et al., 2023a), where our model is trained on progressively longer sequence lengths, starting from 32K tokens and ending at 1M tokens in increasing powers of two. Intuitively, this allows the model to save compute by first learning shorter-range dependencies before moving onto longer sequences. For extending positional embeddings to longer contexts, we adopt a simple, scaled-up version of the approach explored in Rozière et al. (2023), where the $\theta$ parameter for RoPE (Su et al., 2024) is scaled in proportion to the context length. We found this approach to be stable for extending positional embeddings with larger context lengths due to its simplicity, requiring the tuning of only a single hyperparameter. Specifically, we scale the $\theta$ parameter for RoPE alongside increases in context window sizes – the values are shown in Table 6. The progressive training of growing context sizes is shown in Figure 4.

We initialize from LLaMA-2 7B (Touvron et al., 2023b) as base language model and progressively increase the effective context length of the model across 5 stages: 32K, 128K, 256K, 512K, and 1M. For each stage, we train on different filtered versions of the Books3 dataset from The Pile (Gao et al., 2020). Table 6 details information about each training stage, such as the number of tokens, total time, and the Books3 dataset filtering constraints. Each successive run is initialized from the prior sequence length.

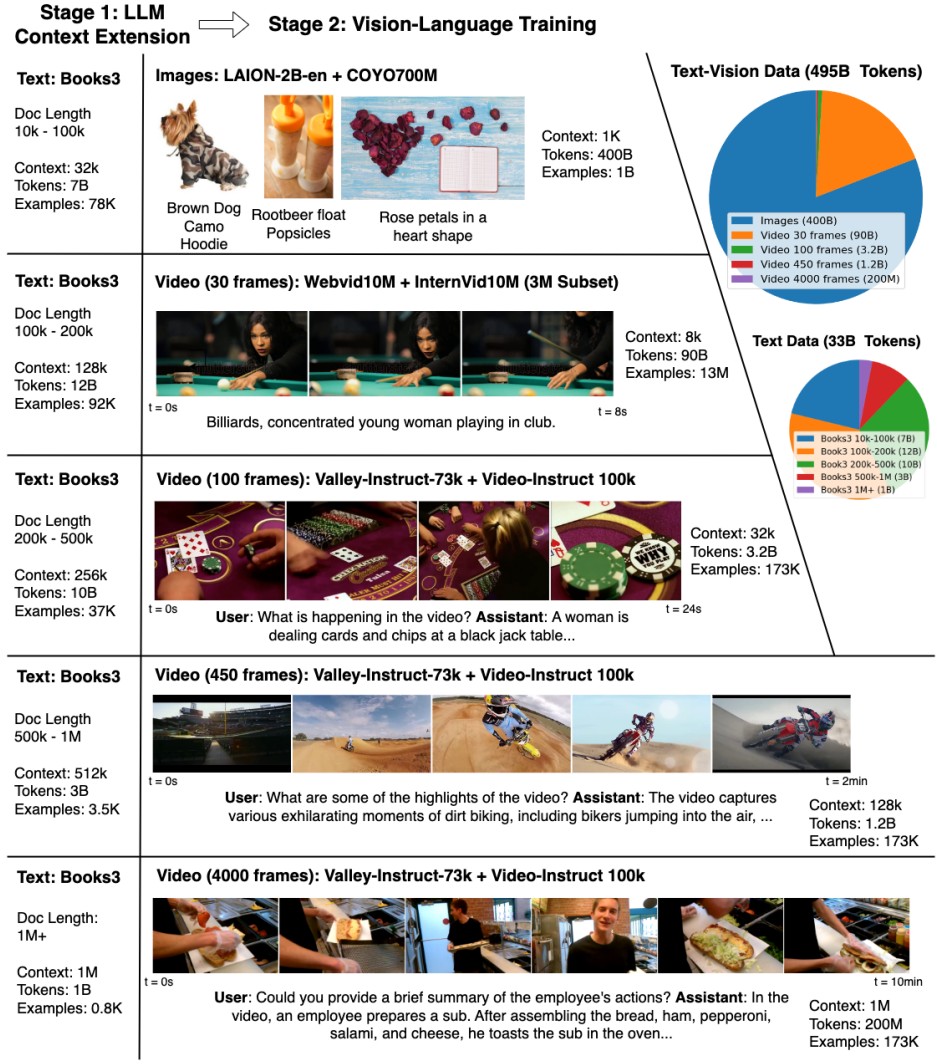

**Figure 4    Curated dataset and training process with progressively increasing data length and complexity**. The diagram outlines a two-stage training process. Stage 1 extends text-based understanding using books datasets of increasing document lengths and token counts. Stage 2 integrates vision-language training. Pie charts display token distribution, showing that images and short-frame videos dominate visual data, while mid-length text examples lead in the text corpus.

## 3.2    MODEL-GENERATED QUESTION-ANSWERING FOR EFFECTIVE CONTEXT

We construct a simple question-answering dataset to develop long-context chat capabilities. First, we split documents from the Books3 dataset into fixed chunks of 1,000 tokens, feed each chunk into our short-context language model, and prompt it to generate a question-answer pair based on the content. To create longer examples (e.g., 32K tokens), we concatenate adjacent chunks and append the relevant question-answer pairs toward the end of the sequence in a chat format. The key intuition is that the model must learn to focus on any part of the context to answer the questions, as the relevant information can appear anywhere within the sequence.

For chat fine-tuning, we train each model on a mix of the UltraChat conversation dataset (Ding et al., 2023) and our custom question-answering dataset, using approximately a 7:3 ratio. We found it crucial to pre-pack the UltraChat data to the training sequence length and keep these examples separate from our question-answering data. This separation is necessary because UltraChat data generally contains a much higher proportion of loss tokens (due to densely packed, short questions in chat), whereas our question-answering data has long questions in chat thus a significantly lower percentage of loss tokens per sequence (< 1%). This difference arises from the long documents in the given context of our question-answering data, which are not included in loss calculations. Table 7

provides further training details for each run. Notably, we do not employ progressive training for any of the chat models; instead, we initialize them from their respective pretrained models at the same context length.

**Summary:** Stage I progressively increase sequence lengths using our curated dataset: starting with 32K tokens and gradually scaling up to 1M tokens. Model-generated question-answering data aids in learning effective long context.

## 3.3 LANGUAGE EVALUATION RESULTS

### 3.3.1 SHORT CONTEXT TASKS

Table 1 presents a comparative analysis between the Llama2-7B model with a 4K context and its context-expanded counterparts, ranging from 32K to 1M. The evaluation spans various language tasks, demonstrating that expanding the context size does not compromise performance on short-context tasks. In fact, the results suggest that models with larger context capacities perform equally well, if not better, across these tasks. This evidence indicates the absence of negative effects from context expansion, highlighting the models' capability to adapt to different task requirements without losing efficiency in shorter contexts.

**Table 1** Performance evaluation across language tasks, comparing Llama-2 7B (4K context window) and context-expanded variants of LWM-Text (32K to 1M). The results demonstrate that increasing context length does not significantly degrade performance on tasks with shorter contexts.

| | | LWM-Text | | | | |
|---|---|---|---|---|---|---|
| **Task / Metric** | **Llama-2 7B** | **32k** | **128k** | **256k** | **512k** | **1M** |
| arc_challenge/acc | 0.40 | 0.43 | 0.45 | 0.44 | 0.44 | 0.43 |
| arc_challenge/acc_norm | 0.43 | 0.47 | 0.47 | 0.46 | 0.46 | 0.46 |
| hellaswag/acc | 0.57 | 0.57 | 0.57 | 0.57 | 0.56 | 0.57 |
| hellaswag/acc_norm | 0.77 | 0.76 | 0.76 | 0.76 | 0.75 | 0.75 |
| mmlu | 0.39 | 0.4 | 0.41 | 0.41 | 0.36 | 0.35 |
| openbookqa/acc | 0.32 | 0.33 | 0.31 | 0.32 | 0.33 | 0.30 |
| openbookqa/acc_norm | 0.44 | 0.44 | 0.44 | 0.43 | 0.41 | 0.41 |

### 3.3.2 RETRIEVAL TASK: SINGLE INFORMATION

We evaluate on the popular Needle In A Haystack task (gkamradt, 2023) – more specifically an version (ArizeAI, 2023) that finds and retrieves random numbers assigned to randomized cities from the context. Figure 2 shows that we can scale to far larger contexts compared to the current best available LLMs. Figure 11 in Appendix shows nearly perfect retrieval accuracy over the entire context of our 1M context model. Appendix C shows more single needle retrieval results for our other shorter context length models.

### 3.3.3 RETRIEVAL TASK: MULTIPLE INFORMATION

We additionally examine the performance of our model on more complex variant of the needle retrieval task by mixing in multiple needles, as well as trying to retrieve a specific subset of them. Figure 5 shows multi-needle retrieval results under different settings. Our model generalizes well when retrieving a single needle from multiple needles in context, with slight degradation when asked to retrieve more than one needle. Table 2 shows multi-needle comparisons, where our model is able to perform competitively or better than GPT-4 at retrieving one needle, or slightly lower performance when retrieving more than one needle. Furthermore, our model is also able to perform well and extend to longer context lengths of up to 1M tokens and far outperforms any recent shorter context baselines applies to longer sequence lengths through positional extrapolation techniques.. However, we note that we see degradation in accuracy while increasing the difficulty of the needle retrieval task, suggesting that there is still more room to improve on the 1M context utilization of our model. We believe that our released model will provide a foundation for future work on developing longer context models, as well as encourage more challenging benchmarks that contain difficult long-range tasks that require higher levels of synthesis, rather than pure fact retrieval.

**Table 2**  Multi Needle in a Haystack. * denotes models **after** the completion of this paper.

| Context Length | Model | $N = 2, R = 2$ | $N = 4, R = 1$ | $N = 4, R = 2$ |
|---|---|---|---|---|
| 32K | Gemini Pro (02/23) | 0.34 | 0.44 | 0.6 |
| | GPT-4-1106 | 0.97 | 0.95 | 0.9 |
| | Llama-3.1-8B-Instruct* | 0.87 | 0.95 | 0.93 |
| | Qwen2.5-7B-Instruct* | **1.0** | **1.0** | **0.97** |
| | Mistral-7B-Instruct-v0.3* | 0.98 | 0.85 | 0.83 |
| | **LWM-Text-1M (Ours)** | 0.84 | 0.97 | 0.84 |
| 128K | Gemini Pro (02/23) | - | - | - |
| | GPT-4-1106 | 0.92 | 0.8 | 0.82 |
| | Llama-3.1-8B-Instruct* | **0.98** | 0.91 | 0.87 |
| | Qwen2.5-7B-Instruct* | **0.98** | 0.80 | **0.90** |
| | Mistral-7B-Instruct-v0.3* | 0.85 | 0.75 | 0.68 |
| | **LWM-Text-1M (Ours)** | 0.83 | **0.98** | 0.83 |
| 1M | Gemini Pro (02/23) | - | - | - |
| | GPT-4-1106 | - | - | - |
| | Llama-3.1-8B-Instruct* | 0.27 | 0.32 | 0.18 |
| | Qwen2.5-7B-Instruct* | 0.0 | 0.0 | 0.0 |
| | Mistral-7B-Instruct-v0.3* | 0.05 | 0.13 | 0.10 |
| | **LWM-Text-1M (Ours)** | **0.67** | **0.84** | **0.69** |

**Figure 5**  Multiple needles retrieval task with LWM-1M. $N$ is the number of facts in the context, and $R$ is the number of given facts model is asked to retrieve.

### 3.3.4  EVALUATION ON LOFT

**Table 3**  Evaluations on some benchmarks in the LOFT dataset.

| Setting: 512K Context | LWM (512K) | GPT-4o (128K) | Claude 3 Opus (200K) |
|---|---|---|---|
| Quora | **0.38** | 0.23 | 0.37 |
| NQ | **0.37** | 0.22 | 0.37 |
| HotPotQA | **0.72** | 0.21 | 0.32 |

We further evaluate our model on a coverage of the LOFT (Lee et al., 2024) dataset collection, we provides a more natural set of benchmarks that examine capabilities for long-context models in the context of document retrieval, and RAG. The benchmark includes tasks such as duplication detection (Quora [1]), document retrieval (HotpotQA (Yang et al., 2018)), and retrieval-based question-answering (NQ). Each dataset contains a corpus of 1000s of documents, and the model is asked to retrieve a set of document ids pertaining to its specific task (Quora, HotpotQA). For RAG (NQ dataset), the model is asked to answer the question using the given context. Table 3 shows evaluations results on 512K context length against various language model baselines.

> **Takeaway:** Long context capability enables LWM to outperform state-of-the-art text models at multiple benchmarks. This demonstrates the effectiveness of our methods for enabling long context.

## 4  STAGE II: EXTENDING TO LONG-CONTEXT VISION-LANGUAGE

Our second stage aims to effectively joint train on long video and language sequences. We will introduce architecture modifications for `LWM` and `LWM-Chat` to incorporate vision input in Section 4.1.

---

[1] https://quoradata.quora.com/First-Quora-Dataset-Release-Question-Pairs

Training on varying sequence lengths is discussed in Section 4.2. The evaluation results are shown in Section 4.3. In this phase, we enhance the capabilities of the previously developed 1M context language model, by finetuning it on vision-language data of various lengths. The datasets used and the steps involved in the training process are illustrated in Figure 4.

## 4.1 ARCHITECTURAL MODIFICATIONS FOR VISION

We use the pretrained VQGAN (Esser et al., 2021) from aMUSEd (Patil et al., 2024) that tokenizes $256 \times 256$ input images to $16 \times 16$ discrete tokens. Videos are tokenized by applying the VQGAN per-frame, and concatenating the codes together. In order to distinguish between modalities when generating, as well as knowing when to switch, we introduce mechanisms to mark the end of text generation / beginning of vision generation, and vice-versa. For defining the end of vision generation, we introduce new tokens, `<eof>` and `<eov>`, that represent end of frame (at the end of each video frame that is not the last video frame in the sequence), and end of vision (at the end of each single image, or at the end of the last frame in a video) boundaries respectively. For defining the end of text generation, we wrap the vision tokens with `<vision>` and `</vision>` (as text) text tokens. The model is trained with interleaved concatenations of vision and text tokens, and predicted autoregressively (see Figure 3).

## 4.2 TRAINING STEPS

We initialize from our LWM-Text-1M text model, and perform a similar process of progressive training on a large amount of combined text-image and text-video data, with the exception that we do not additionally scale RoPE $\theta$, as it already supports up to 1M context. Table 8 shows details for each training stage, where the model is initialized from the prior shorter sequence length stage. For each stage, we train on the following data:

- `LWM-1K`: We train on large set of text-image dataset comprising of a mix of LAION-2B-en (Schuhmann et al., 2022) and COYO-700M (Byeon et al., 2022). The datasets were filtered to only include images with at least 256 resolution – in total roughly 1B text-image pairs. During training, we concatenate the text-image pairs and randomly swap the order of the modalities to model both text-image generation, unconditional image generation, and image captioning. We pack text-image pairs to sequences of 1K tokens.

- `LWM-8K`: We train on a text-video dataset mix of WebVid10M (Bain et al., 2021) and 3M Intern-Vid10M (Wang et al., 2023) examples. Similar to prior works (Ho et al., 2022a;b; Villegas et al., 2022), we jointly train on both images and video with a 50-50 ratio of each modality. We pack images to sequences of 8K tokens, and 30 frame videos at 4FPS. Similar to image training, we randomly swap the order of modalities for each text-video pair.

- `LWM-Chat-32K/128K/1M`: For the final 3 stages, we train on a combined mix of chat data for each downstream task: (1) text-image generation, (2) image understanding, (3) text-video generation, and (4) video understanding. We construct a simple version of text-image and text-video chat data by sampling random subsets of the pretraining data augmented with chat format. For image understanding, we using the image chat instruct data from ShareGPT4V (Chen et al., 2023a). Lastly, for the video understanding chat data, we use a combined mix of Valley-Instruct-73K (Luo et al., 2023) and Video-ChatGPT-100K instruct data (Maaz et al., 2023). For all short context data (image generation, image understanding, video generation), we pack sequences to the training context length. During packing, we found it crucial to mask out the attention so that each text-vision pair only attends to itself, as well as re-weighting losses to make computation identical to training in a non-packed + padding training regime. For video understanding data, we uniformly sample a max number of frames to fit the training context length of the model if the video is too long. During training, We allocate 25% of each batch to each of the 4 downstream tasks.

For the first two stages of training (LWM-1K and LWM-8K), we additionally mix 16% of the batch to be pure text data from OpenLLaMA (Geng and Liu, 2023), as we found it beneficial to preserve language capabilities while training on vision data.

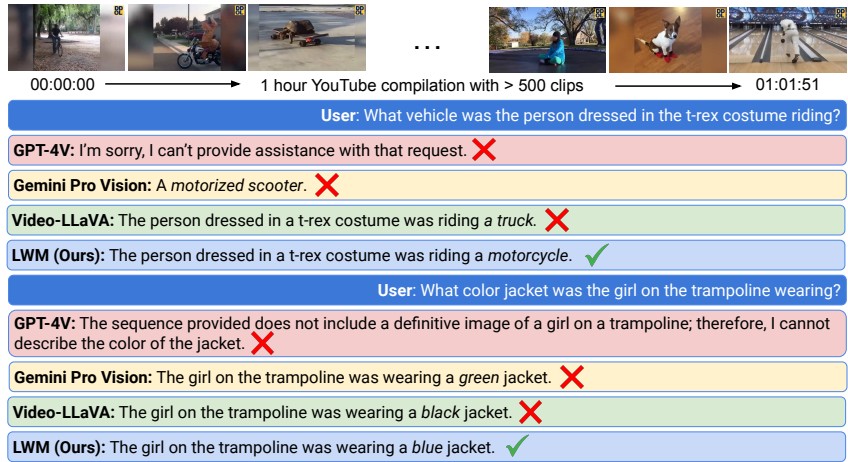

**Figure 6   LWM excels in answering questions about a 1-hour YouTube video**. This figure compares LWM-Chat-1M with proprietary models like Gemini Pro Vision and GPT-4V, along with open-source models. The test involves answering questions based on an hour-long YouTube compilation containing over 500 video clips. LWM demonstrates superior performance in providing accurate answers requiring comprehension of extended video content.

**Table 4**   Long Video-MME Benchmark. * denotes models **after** the completion of this paper.

| Method | Parameters | Frames | Medium (4min-15min) | Long (30min-60min) |
|---|---|---|---|---|
| Gemini 1.5 Pro* | Unknown | ≤ 1800 | 74.3 | 67.4 |
| GPT-4o* | Unknown | 384 | 70.3 | 65.3 |
| LLaVA-Video* | 72B | 64 | 68.9 | 61.5 |
| VideoLLaMA 2* | 72B | 32 | 59.9 | 57.6 |
| Long-LLaVA* | 7B | 64 | 51.4 | 45.4 |
| Video-LLaVA | 7B | 8 | 38.1 | 36.2 |
| LWM-1M | 7B | ≤ 1800 | 63.7 | 60.8 |

**Summary:** Stage II training incorporates image and video. Building on Stage I, it gradually increases sequence lengths of vision and text input. Importantly, we found our masked sequence packing and mixing synthetic and chat data crucial to retain short context performance during our progressive training. Appendix B shows ablations when not using our training method on instruction-following and text-image understanding benchmarks.

## 4.3   VISION-LANGUAGE EVALUATION RESULTS

### 4.3.1   LONG VIDEO UNDERSTANDING

Although vision-language model (Lin et al., 2023; OpenAI, 2023; Team et al., 2023) can ingest long videos, this is commonly done by performing large temporal subsampling of video frames due to limited context length. For example, Video-LLaVA (Lin et al., 2023) is restricted to uniformly sampling 8 frames from a video, no matter how long the original video may be. As such, models may lose more fine-grained temporal information that is important for accurately answering any questions about the video. In contrast, our model is trained on long sequences of 1M tokens, and as a result, can simultaneously attend thousands of frames of videos to retrieve fine-grained information over short time intervals. Table 4 shows long video evaluations on the Video-MME (Fu et al., 2024) benchmark, demonstrating our model as the best performing model among its size class. Figure 6 shows an example of our model correctly answering questions about a long, 1-hour YouTube compilation consisting of more than 500 individual clips. Our baseline methods, on the other hand, generally have difficulty answering the questions due to a limited number of frames. More results are shown in Figure 18 and Appendix F.

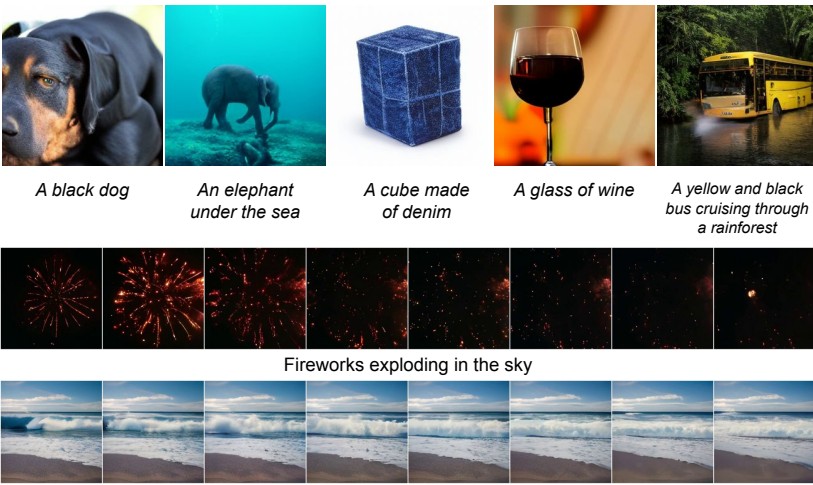

*A black dog*     *An elephant under the sea*     *A cube made of denim*     *A glass of wine*     *A yellow and black bus cruising through a rainforest*

Fireworks exploding in the sky

Waves crashing against the shore

**Figure 7** LWM's ability to generate both static images and dynamic videos from text is shown. The top row illustrates image, while the bottom rows show video.

### 4.3.2 IMAGE UNDERSTANDING AND SHORT VIDEO UNDERSTANDING

We evaluate LWM on standard benchmarks for image and short video understanding, with results presented in Table 5. Our model performs comparably to baselines but falls short of state-of-the-art (SOTA) models. This performance gap is not unexpected, given that SOTA models leverage vision backbones that have undergone extensive CLIP training (Radford et al., 2021). In contrast, LWM utilizes discrete tokens from an off-the-shelf model (Patil et al., 2024). Discrete tokens result in greater information loss, particularly for OCR-like textual data, compared to continuous CLIP embeddings. Moreover, our model learns text-image alignment from scratch, while CLIP-based models benefit from large-scale pretraining. This work primarily focuses on long-context methodology, and we defer additional training to future work due to computational constraints. A straightforward approach to improving benchmark scores would be to incorporate CLIP embeddings as additional input. Despite not achieving SOTA scores on these short video benchmarks, we believe LWM provides valuable insights for future long-context language and video understanding and generation. The model's performance could be enhanced through additional training and minor modifications. We include qualitative image understanding examples in Appendix E and qualitative video understanding examples in Appendix F.

### 4.3.3 IMAGE AND VIDEO GENERATION

Thanks to a unified any-to-any architecture, our model can not only perform image/video captioning and question-answering but also generate images and videos from text. Figure 7 demonstrates examples of these capabilities. For autoregressive sampling, we employ classifier-free guidance (Ho and Salimans, 2022) on the logits, similar to previous works (Yu et al., 2022; Gafni et al., 2022). In the unconditional branch, we initialize each sequence with `<bos><vision>`. For additional image and video generation examples, please refer to Appendices H and I, respectively.

> **Takeaway:** LWM excels in long video understanding by processing significantly more frames than previous state-of-the-arts, resulting in better understanding. Moreover, its long-context enabled unified any-to-any architecture allows for versatile image and video and text understanding and generation.

**Table 5** Image Understanding Benchmarks (left) and Video Understanding Benchmarks (right)

| Method | Visual Token | VQAv2 | GQA | SQA | Method | MSVD | MSRVTT | TGIF |
|---|---|---|---|---|---|---|---|---|
| MiniGPT-4 | CLIP | - | 30.8 | 25.4 | VideoChat | 56.3 | 45 | 34.4 |
| Otter | CLIP | - | 38.1 | 27.2 | LLaMA-Adapte | 54.9 | 43.8 | - |
| InstructBLIP | CLIP | - | 49.2 | 60.5 | Video-LLaMA | 51.6 | 29.6 | - |
| LLaVA-1.5 | CLIP | 78.5 | 62.0 | 66.8 | Video-ChatGPT | 64.9 | 49.3 | 51.4 |
| LWM (ours) | VQGAN | 55.8 | 44.8 | 47.7 | LWM (ours) | 55.9 | 44.1 | 40.9 |

## 5 RELATED WORKS

Our research builds upon existing efforts to extend the context windows of language models, enabling them to process more tokens (Chen et al., 2023b; Tworkowski et al., 2023; Liu et al., 2023c). These approaches often employ innovative extrapolation techniques to expand pretrained positional encodings, followed by model finetuning on longer context data. In contrast, our model takes a straightforward approach by incrementally increasing $\theta$ in RoPE positional encodings alongside expanding the training context window sizes, which we found to be effective. Additionally, there have been investigations into architectures that avoid modeling pairwise interactions, such as sparse attention and sliding window techniques (Child et al., 2019; Beltagy et al., 2020). Prior research has explored sequence parallelization (Li et al., 2021; Korthikanti et al., 2022, inter alia), though it is not optimized for blockwise transformers or compatible with memory-efficient attention, both of which are critical for large context training. Our work further leverages large context transformer techniques (Liu et al., 2024; Liu and Abbeel, 2023) to capture exact pairwise interactions in extended sequences for enhanced performance. Load-balancing strategies, such as skipping causal masked computation (Brandon et al., 2023; Li et al., 2023) offer room for further optimization. Concurrent developments like Gemini 1.5 (Reid et al., 2024) reach 1M tokens context size in language and video.

Additionally, our approach relates closely to advances in instruction tuning (Taori et al., 2023; Chiang et al., 2023; Geng et al., 2023), which focus on finetuning models with conversational data to boost their performance across diverse language tasks. We aim to extend these capabilities to the domain of long-sequence understanding in both video and language tasks. To achieve this, we extend the model's context size by training on comprehensive datasets, including books and long videos, and finetune on model-generated question-answering datasets to enhance its ability to handle extended conversational sequences.

Furthermore, our research draws from work on integrating vision capabilities into language models (Liu et al., 2023b; Lin et al., 2023; Awadalla et al., 2023; Zhang et al., 2023; Jin et al., 2023b; Aiello et al., 2023). These efforts frequently utilize continuous embeddings (Radford et al., 2021; Li et al., 2022) to encode visual information into embeddings for inputting into language models. While these approaches benefit from CLIP's cross-modal understanding to encode textual information from images, their ability to predict text from visual input is limited, as is their capacity to learn from diverse visual-language formats. In contrast, our autoregressive model, which processes "tokens in, tokens out," allows greater flexibility in modeling various formats, including image-text, text-image, text-video, video-text, and pure formats like video, image, or text. Our method is compatible with these prior works, making it an interesting future direction to combine continuous embeddings as input with discrete tokens and a long-context autoregressive model.

## 6 CONCLUSION

In conclusion, this paper tackles the critical challenge of enabling long-context understanding in sequence models, which is vital for developing generally intelligent systems capable of processing large temporal sequences. By exploring the development of 1M context language and video-language models, the work sets new benchmarks in language retrieval and long video understanding. We outline approaches to data curation and progressive context extension, accompanied by an efficient open-source implementation for scalable training on long sequences. Moreover, we open-source a family of 7B parameter models capable of handling over 1M tokens in text and video.

**Limitations**. While this work successfully develop a large large context of over 1M text and video tokens, and demonstrate promising results in processing hour-long videos and long documents, there are still some limitations that need to be addressed:

- Improved tokenization and embedding. This work uses a vanilla image tokenizer for images and frame-by-frame tokenization for videos. Future work could explore video tokenization that takes time redundancy into account, as well as including continuous embeddings as input to enrich image understanding.
- Limited scale. Our models use more tokens per parameter than Chinchilla's recommendation, but being much smaller than current large language models (100B+ parameters), our findings may not directly apply to them. Extrapolating to larger scales should be done cautiously, as different scaling behaviors could emerge at those larger sizes.

## ACKNOWLEDGMENTS

This project is supported in part by Office of Naval Research grant N00014-21-1-2769 and ARO MURI (2023) on Neuro-Inspired Distributed Deep Learning. We thank Google TPU Research Cloud for granting us access to TPUs, and thank Google Cloud for granting us research credits for storage. Pieter Abbeel holds concurrent appointments as a Professor at UC Berkeley and as an Amazon Scholar. This paper describes work performed at UC Berkeley and is not associated with Amazon.

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

## A  FURTHER DETAILS

**Model Flops Utilization**.  We trained our models using TPUv4-1024, which is approximately equivalent to 450 A100s, with a batch size of 8M using FSDP (Facebook, 2023) and Blockwis-eRingAttention (Liu et al., 2024) for large contexts. Figure 8 shows the model FLOPS utilization (MFU) for each training stage. Blue color bars show language training and orange color bars show vision-language training. Our training achieves good MFUs even for very large context sizes.

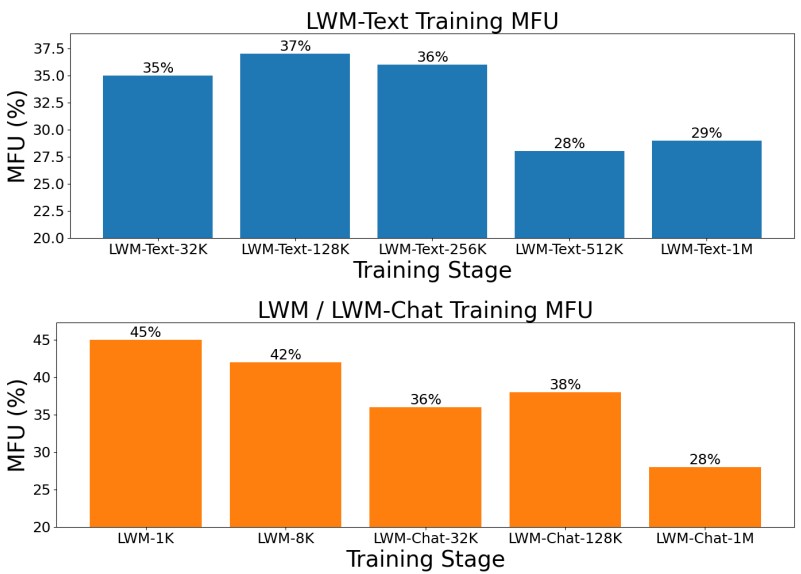

**Figure 8  High MFU training across sequence lengths.** Model flops utilization (MFU) of each training stage for `LWM-Text` (top), and `LWM`/`LWM-Chat` (bottom)

**Training Loss Curves**. Figure 9 and Figure 10 show the training loss curves for each stage of training the language and vision-language models respectively.

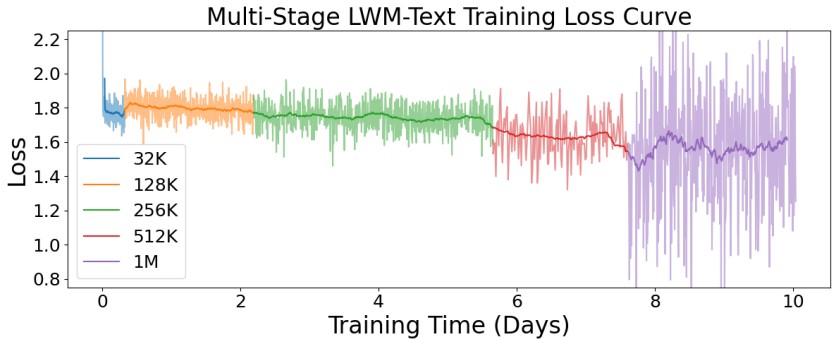

**Figure 9  Training progress over multiple days for `LWM-Text`.** Train loss curve for each training stage for `LWM-Text` models.

**Training Hyperparameters**. See Appendix **??**

**Scaling Inference**. We additionally scale our inference code to support million-length sequences by implementing RingAttention for decoding. Inference for such long sequences requires a minimum of v4-128 with a TPU mesh sharding of 32 tensor parallelism, and 4 sequence parallelism (ring dimension). We perform inference in pure single precision, where additional improvements can be made through techniques in scalability such as quantization.

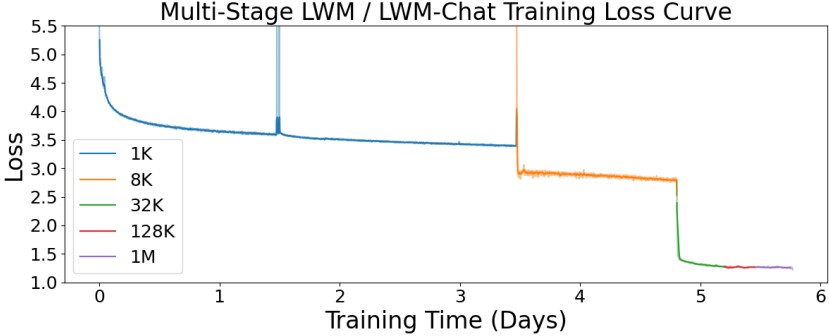

**Figure 10** `Training progress over multiple days for LWM`. Train loss curve for each training stage for LWM and LWM-Chat models. Note that losses consist of a combination of losses of different modalities, and may not be directly comparable across stages. The sharp peak in the middle of 1K training is due to newly incporating EOF and EOV tokens into the vision codebook.

**Table 6**   LWM-Text Training Stages

|  | 32K | 128K | 256K | 512K | 1M |
|---|---|---|---|---|---|
| Parameters | 7B | 7B | 7B | 7B | 7B |
| Sequence Length | $2^{15}$ | $2^{17}$ | $2^{18}$ | $2^{19}$ | $2^{20}$ |
| RoPE $\theta$ | 1M | 10M | 10M | 25M | 50M |
| Tokens per Batch | 4M | 4M | 4M | 4M | 4M |
| Total Tokens | 4.8B | 12B | 12B | 3B | 1.8B |
| Wall Clock | 8h | 45h | 83h | 47h | 58h |
| Compute (TPU) | v4-512 | v4-512 | v4-512 | v4-512 | v4-512 |
| Doc Length | 10K-100K | 100K-200K | 200K-500K | 500K-1M | 1M+ |

**Table 7**   LWM-Text-Chat Training Details

|  | 128K | 256K | 512K | 1M |
|---|---|---|---|---|
| Parameters | 7B | 7B | 7B | 7B |
| Sequence Length | $2^{17}$ | $2^{18}$ | $2^{19}$ | $2^{20}$ |
| RoPE $\theta$ | 10M | 10M | 25M | 50M |
| Tokens per Batch | 4M | 4M | 4M | 4M |
| Total Tokens | 1.2B | 1.2B | 1.2B | 1.2B |
| Wall Clock | 6h | 10h | 20h | 40h |
| Compute (TPU) | v4-512 | v4-512 | v4-512 | v4-512 |

**Table 8**   LWM and LWM-Chat Training Stages

|  | 1K | 8K | Chat-32K | Chat-128K | Chat-1M |
|---|---|---|---|---|---|
| Parameters | 7B | 7B | 7B | 7B | 7B |
| Sequence Length | $2^{10}$ | $2^{13}$ | $2^{15}$ | $2^{17}$ | $2^{20}$ |
| RoPE $\theta$ | 50M | 50M | 50M | 50M | 50M |
| Tokens per Batch | 8M | 8M | 8M | 8M | 8M |
| Total Tokens | 363B | 107B | 10B | 3.5B | 0.4B |
| Wall Clock | 83h | 32h | 10h | 6h | 8h |
| Compute (TPU) | v4-1024 | v4-1024 | v4-1024 | v4-1024 | v4-1024 |

# B ABLATION STUDIES

## B.1 MASKED SEQUENCE PACKING

As mentioned in Section 4.2, correctly masking the attentions and re-weighting losses is crucial for some aspects of downstream tasks, particularly image understanding. Table 9 shows a comparison of our model with and without packing corrections. Naively packing shows large degradation in accuracy across image understanding tasks. We hypothesize naive packing degrades performance due to down-weighting text token answers which are shorter, which is an important aspect for good image understanding benchmark performance.

**Table 9** Ablation study comparing standard independent packing and our masked sequence packing mechanisms across three tasks. Results show that masked sequence packing significantly improves performance across all tasks.

|  | VQAv2 | SQA | POPE |
|---|---|---|---|
| Standard independent packing | 48.3 | 34.8 | 62.5 |
| Masked sequence packing (Ours) | **55.8** | **47.7** | **75.2** |

## B.2 MIXING SYNTHETIC AND CHAT DATA

We additionally evaluate the our model on MT-Bench (Zheng et al., 2023) to test its conversation ability. Table 10 shows the MT-Bench scores of for each of our models. Table 11 illustrates the relationship between the mix of chat and fact retrieval tasks and the performance on MT-Bench score and Needle Retrieval accuracy. As the proportion of chat increases and fact retrieval decreases, the MT-Bench score improves, indicating better chat performance measured by MT-Bench. Conversely, Needle Retrieval accuracy decreases, suggesting a trade-off where increasing chat interaction capabilities may reduce the system's precision in retrieving specific information or 'needles' from input context. Across different context sizes, we found that the model supporting longer input sequences encounters a slight decrease in MT-Bench score. We hypothesize that this is because we chose to train with fewer examples on longer sequence training and can be improved by simply training on more data. In addition, this trade-off may be resolved by acquiring higher quality long-context chat data that is closer to the chat distribution of the UltraChat dataset.

**Table 10** Results on MT-Bench across different context sizes. Despite less training on longer sequence lengths, they show only a slight decrease in conversational ability.

| Model | MT-Bench |
|---|---|
| LWM-Text-Chat-128k | 4.62 |
| LWM-Text-Chat-256k | 5 |
| LWM-Text-Chat-512k | 4.83 |
| LWM-Text-Chat-1M | 4.19 |

**Table 11** Relationship between the mix of chat and fact retrieval tasks and the performance on MT-Bench score and Needle Retrieval accuracy.

| Chat / QA Mix | MT-Bench | Needle Acc |
|---|---|---|
| 0% / 100% | 2.42 | 100% |
| 40% / 60% | 4.14 | 100% |
| 70% / 30% | 4.62 | 96% |
| 90% / 10% | 5.1 | 55% |
| 100% / 0% | 5.8 | 31% |

# C   MORE SINGLE-NEEDLE RETRIEVAL RESULTS

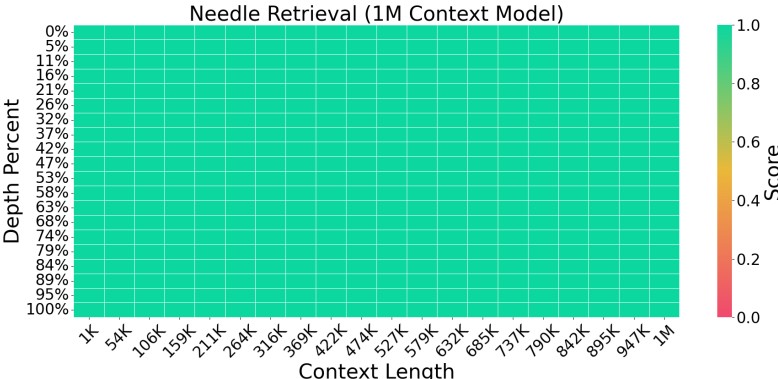

**Figure 11**   Needle retrieval task using the LWM-Text-Chat-1M model. The model demonstrates near-perfect retrieval accuracy across various positions within the 1M context window, as reflected by consistently high scores at different depth percentages and context lengths.

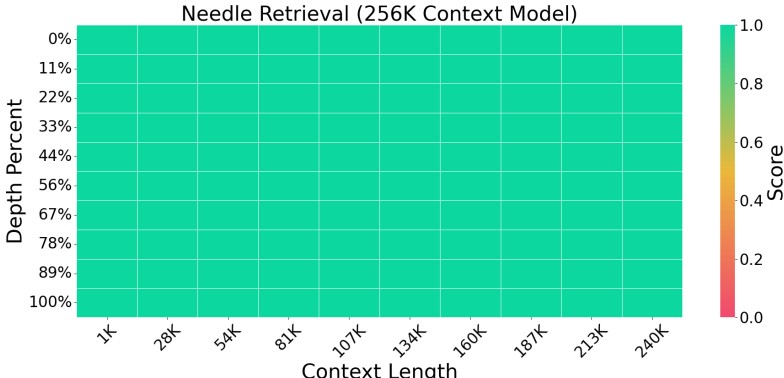

**Figure 12**   Single needle retrieval accuracy for the LWM-Text-Chat-256K model. The model achieves near-perfect retrieval performance across various positions in the 256K context window, as shown by consistently high scores across all depth percentages and context lengths.

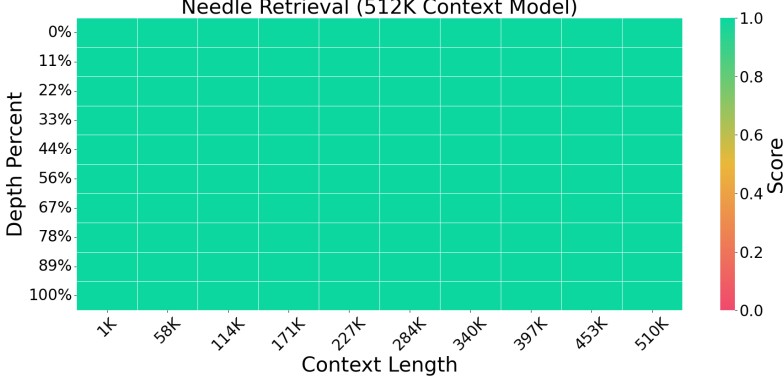

**Figure 13**   Single needle retrieval accuracy for the LWM-Text-Chat-512K model. The model demonstrates near-perfect retrieval performance across different positions within the 512K context window, as indicated by consistently high scores at varying depth percentages and context lengths.

## D    DETAILS OF THE MULTI-NEEDLE EVALUATIONS

We built on top of the original Needle in a Haystack problem (see Section 3.3.2 for details), which was to retrieve a single magic number randomly inserted into some long context. In this case, magic numbers are associated with random cities ("The magic number for San Francisco is 2521233" → "What is the magic number for San Francisco?"). We extend the task by introducing two variables N and R, where N is the number of needles (magic numbers + different cities) randomly inserted into the context, and R is the random subset of magic numbers asked to retrieve ("What are the magic numbers for San Francisco, . . . , and Bangkok?"). Correctness is computed by extracting out the numbers retrieved for each cities and checked with string matching.

## E    MORE IMAGE UNDERSTANDING EXAMPLES

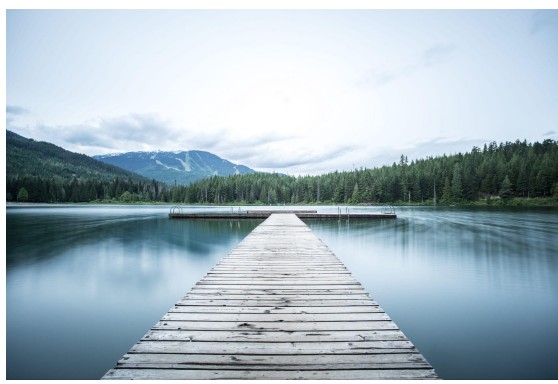

**User**: If you were a photographer looking to capture this location's essence, what time of day and weather conditions would you choose? Describe the reasons behind your choice.

**Assistant**: I would choose the time of day to be during sunset. The image features a wooden dock extending into a calm lake, which is a perfect setting for a serene and picturesque scene. The sunset casts a warm, golden light on the lake, enhancing the tranquility of the scene. Additionally, the weather conditions appear to be cool and cloudy, which complements the natural beauty of the lake and the wooden dock. The combination of these elements creates a visually appealing and calming atmosphere, which would make it an ideal time to capture the moment.

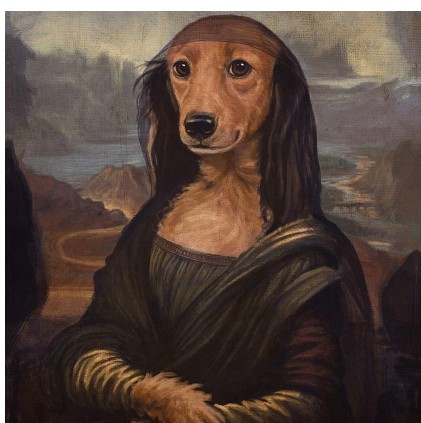

**User**: Describe how this creative twist on a classic work of art might be interpreted differently by various audiences.

**Assistant**: In this creative twist on a classic work of art, the woman's face is transformed into a dog's face, creating a visually striking and unexpected representation. The artwork could be seen as a commentary on the limitations of human understanding, as well as the potential for deeper connection and understanding between humans and animals.

**Figure 14**    Question answering based on image input using LWM. The assistant provides detailed responses to questions about capturing the essence of a serene lakeside scene and interpreting a creative twist on a classic artwork featuring a dog.

## F    MORE VIDEO UNDERSTANDING EXAMPLES

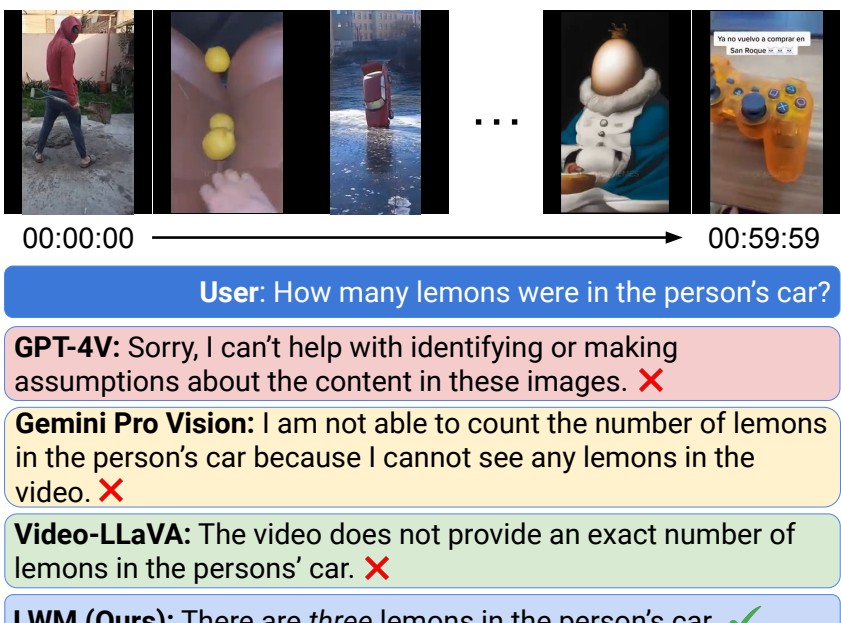

**Figure 15**   LWM demonstrates video understanding over a 1-hour video. The figure shows a comparison of different AI systems responding to a question about the number of lemons in a person's car. While GPT-4V, Gemini Pro Vision, and Video-LLAVA fail to provide a correct response, LWM accurately identifies that there are three lemons in the car.

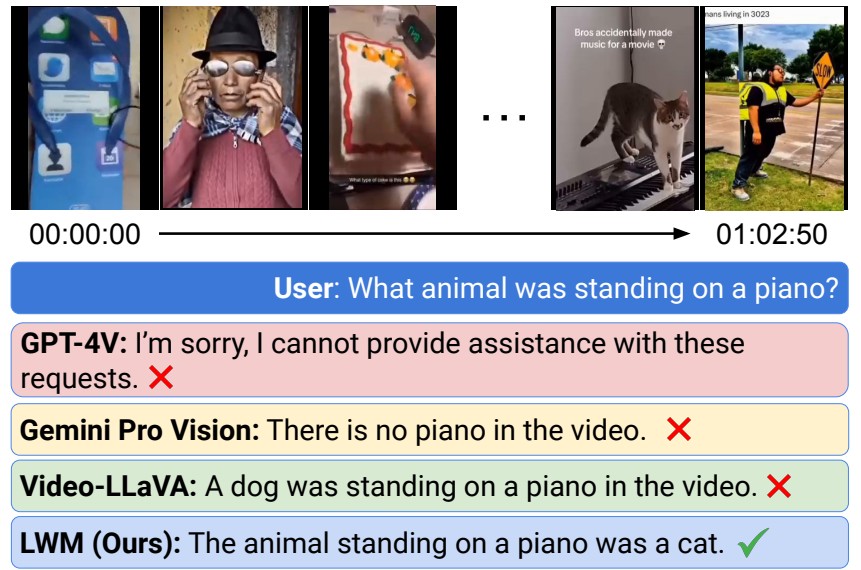

**Figure 16**   LWM demonstrates video understanding over a 1-hour video. The figure compares AI systems' responses to the question 'What animal was standing on a piano?' While GPT-4V, Gemini Pro Vision, and Video-LLAVA provide incorrect or incomplete answers, LWM correctly identifies that the animal standing on the piano was a cat.

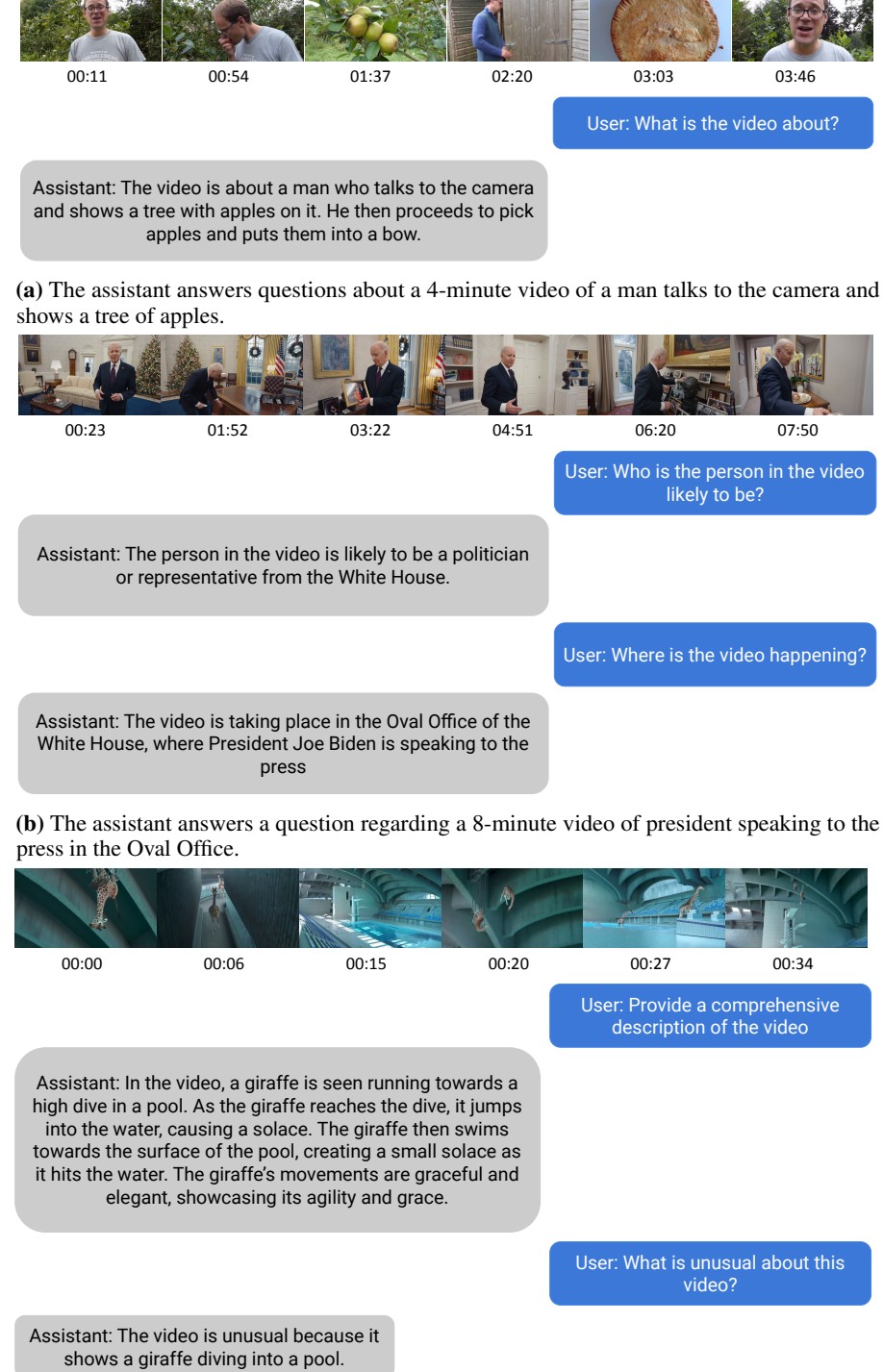

**(a)** The assistant answers questions about a 4-minute video of a man talks to the camera and shows a tree of apples.

**(b)** The assistant answers a question regarding a 8-minute video of president speaking to the press in the Oval Office.

**(c)** The assistant answers a question about a 30-second video of a giraffe.

**Figure 17** Answering questions about videos using LWM. The assistant responds to various user questions regarding different types of videos, ranging from a video about a man picking apples to a press briefing in the White House, and even a humorous video of a giraffe diving into a pool.

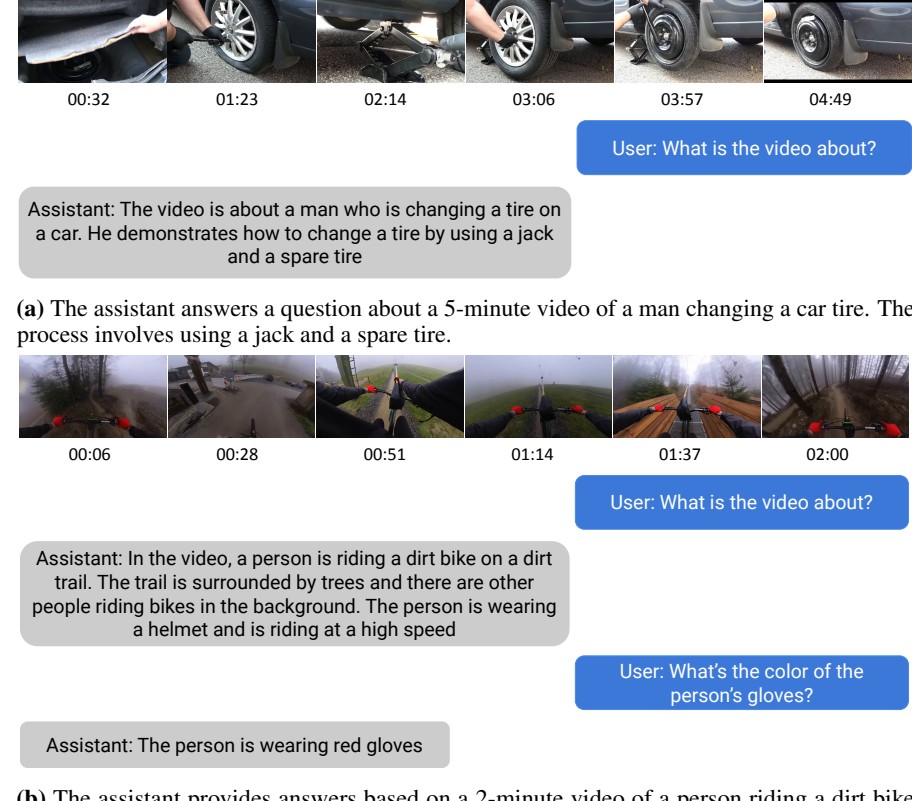

**(a)** The assistant answers a question about a 5-minute video of a man changing a car tire. The process involves using a jack and a spare tire.

**(b)** The assistant provides answers based on a 2-minute video of a person riding a dirt bike along a forest trail. The rider wears a helmet and red gloves, traveling at high speed.

**Figure 18**    The system (LWM) successfully answers questions about video content.

## G    DETAILS OF QUALITATIVE VIDEO UNDERSTANDING EVALUATION

For qualitative evaluation of our videos, we source various videos from YouTube that cover a range of topics, such as ego-centric camera, how to videos, interviews, and animations. We evaluate all videos at 1FPS, and sample uniformly a max number of frames for videos that are longer than what our video can support at 1 FPS. Videos are additionally resized and center cropped to $256 \times 256$ resolution before inputting into the model.

## H  MORE IMAGE GENERATION EXAMPLES

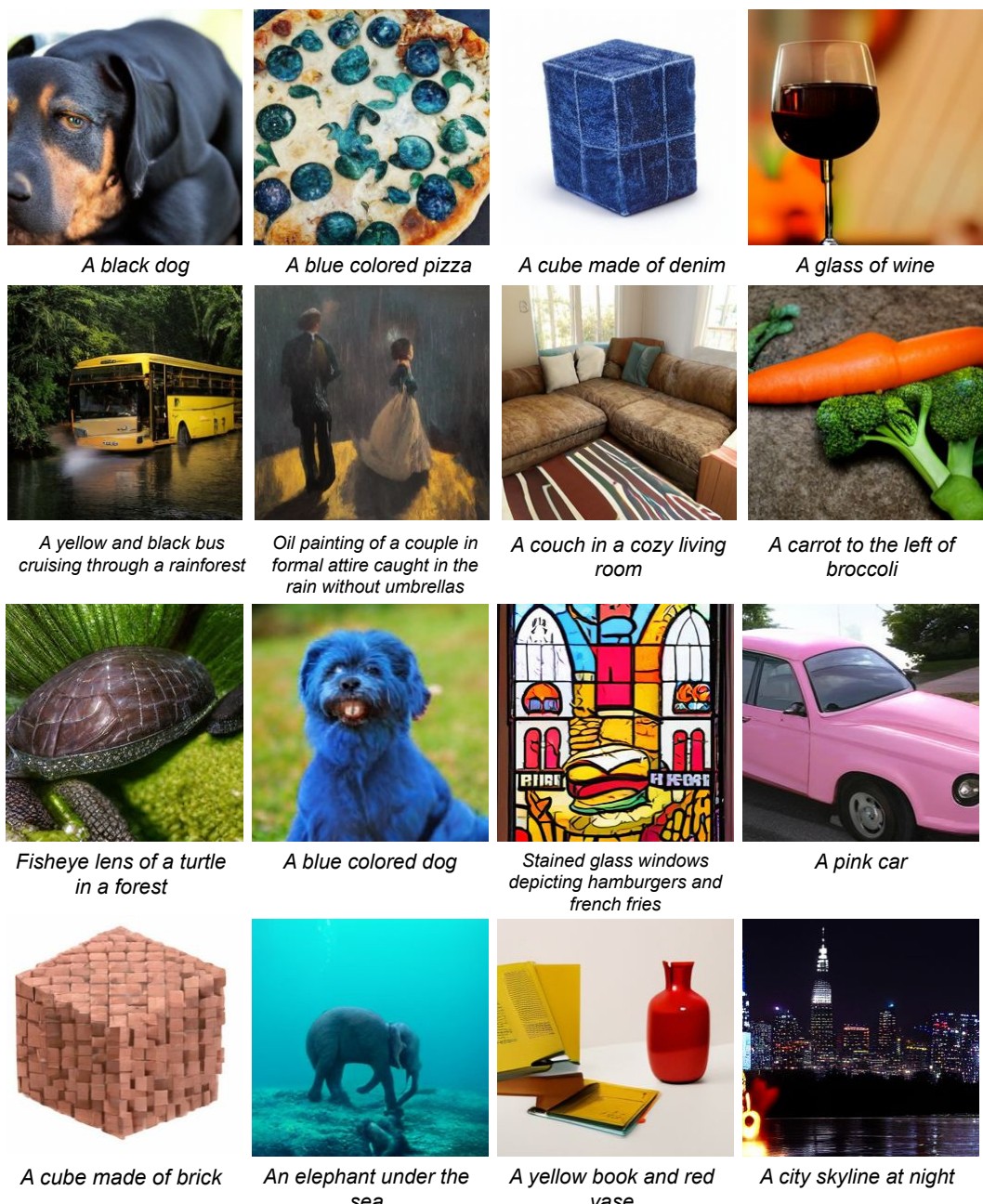

**Figure 19**  Images generation using LWM, showcasing various scenes and objects.

## I   MORE VIDEO GENERATION EXAMPLES

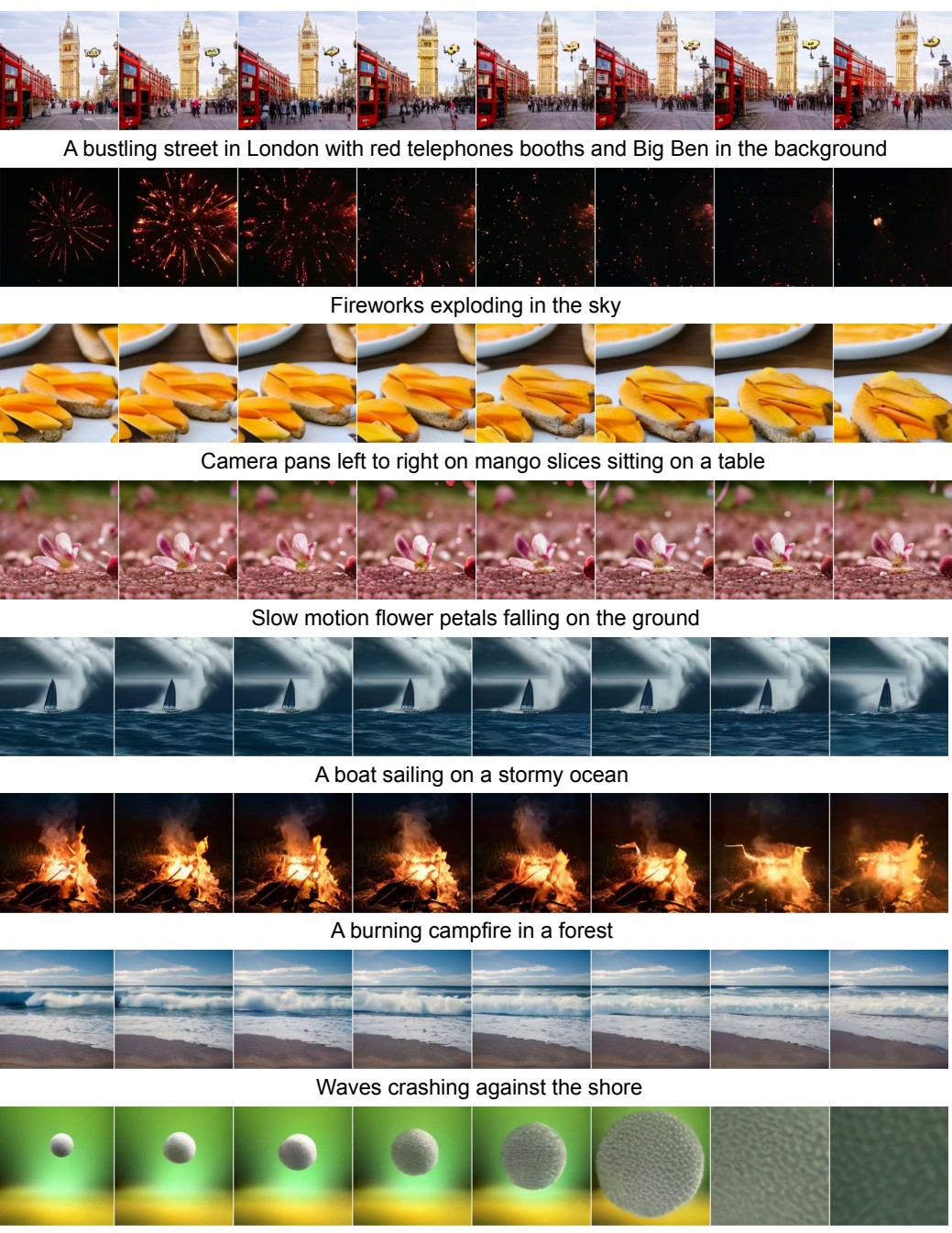

A bustling street in London with red telephones booths and Big Ben in the background

Fireworks exploding in the sky

Camera pans left to right on mango slices sitting on a table

Slow motion flower petals falling on the ground

A boat sailing on a stormy ocean

A burning campfire in a forest

Waves crashing against the shore

A ball thrown in the air

**Figure 20**   Video sequences generated using LWM, showing various scenes.

## J  TRAINING HYPERPARAMETERS

**Table 12**  LWM-Text Training Stages

|  | 32K | 128K | 256K | 512K | 1M |
|---|---|---|---|---|---|
| Parameters | 7B | 7B | 7B | 7B | 7B |
| Initialize From | LLaMA-2 7B | Text-32K | Text-128K | Text-256K | Text-512K |
| Precision | float32 | float32 | float32 | float32 | float32 |
| Sequence Length | $2^{15}$ | $2^{17}$ | $2^{18}$ | $2^{19}$ | $2^{20}$ |
| RoPE $\theta$ | 1M | 10M | 10M | 25M | 50M |
| Tokens per Batch | 4M | 4M | 4M | 4M | 4M |
| Total Tokens | 4.8B | 12B | 12B | 3B | 1.8B |
| Total Steps | 1200 | 3000 | 3000 | 720 | 450 |
| LR Schedule | Constant | Constant | Constant | Constant | Constant |
| LR Warmup Steps | 100 | 200 | 200 | 50 | 25 |
| LR | $4 \times 10^{-5}$ | $4 \times 10^{-5}$ | $4 \times 10^{-5}$ | $4 \times 10^{-5}$ | $4 \times 10^{-5}$ |
| Compute (TPU) | v4-512 | v4-512 | v4-512 | v4-512 | v4-512 |
| Mesh Sharding | 1,-1,4,1 | 1,-1,8,1 | 1,-1,16,1 | 1,-1,16,2 | 1,-1,16,4 |

**Table 13**  LWM-Text-Chat Training Details

|  | 128K | 256K | 512K | 1M |
|---|---|---|---|---|
| Parameters | 7B | 7B | 7B | 7B |
| Initialize From | Text-128K | Text-256K | Text-512K | Text-1M |
| Precision | float32 | float32 | float32 | float32 |
| Sequence Length | $2^{17}$ | $2^{18}$ | $2^{19}$ | $2^{20}$ |
| RoPE $\theta$ | 10M | 10M | 25M | 50M |
| Tokens per Batch | 4M | 4M | 4M | 4M |
| Total Tokens | 1.2B | 1.2B | 1.2B | 1.2B |
| Total Steps | 300 | 300 | 300 | 300 |
| LR Schedule | Constant | Constant | Constant | Constant |
| LR Warmup Steps | 25 | 25 | 25 | 25 |
| LR | $4 \times 10^{-5}$ | $4 \times 10^{-5}$ | $4 \times 10^{-5}$ | $4 \times 10^{-5}$ |
| Compute (TPU) | v4-512 | v4-512 | v4-512 | v4-512 |
| Mesh Sharding | 1,-1,4,1 | 1,-1,8,1 | 1,-1,16,1 | 1,-1,16,2 |

**Table 14**  LWM / LWM-Chat Training Stages

|  | 1K | 8K | 32K | 128K | 1M |
|---|---|---|---|---|---|
| Parameters | 7B | 7B | 7B | 7B | 7B |
| Initialize From | Text-1M | 1K | 8K | 32K | 128K |
| Precision | float32 | float32 | float32 | float32 | float32 |
| Sequence Length | $2^{10}$ | $2^{13}$ | $2^{15}$ | $2^{17}$ | $2^{20}$ |
| RoPE $\theta$ | 50M | 50M | 50M | 50M | 50M |
| Tokens per Batch | 8M | 8M | 8M | 8M | 8M |
| Total Tokens | 363B | 107B | 10B | 3.5B | 0.4B |
| Total Steps | 45000 | 14000 | 1200 | 450 | 50 |
| LR Schedule | Cosine | Cosine | Cosine | Cosine | Cosine |
| LR Warmup Steps | 1000 | 500 | 100 | 50 | 5 |
| Max LR | $6 \times 10^{-4}$ | $6 \times 10^{-4}$ | $8 \times 10^{-5}$ | $8 \times 10^{-5}$ | $8 \times 10^{-5}$ |
| Min LR | $6 \times 10^{-5}$ | $6 \times 10^{-5}$ | $8 \times 10^{-5}$ | $8 \times 10^{-5}$ | $8 \times 10^{-5}$ |
| Compute (TPU) | v4-1024 | v4-1024 | v4-1024 | v4-1024 | v4-1024 |
| Mesh Sharding | 1,-1,1,1 | 1,-1,1,1 | 1.-1.4,1 | 1.-1.8,1 | 1,-1,16,4 |

