# OpenReview forum: "World Model on Million-Length Video And Language With Blockwise RingAttention"
_ICLR.cc/2025/Conference — ICLR 2025 Poster_

### Official Review · Reviewer_cWVz · 2024-10-16

**Soundness:** 4
**Presentation:** 3
**Contribution:** 4
**Rating:** 6
**Confidence:** 5

**Summary:**

The paper scales the LLM context length up to 1M tokens by utilizing Blockwise RingAttention. They progressively increase the context length from 32K to 1M using book-length text data and video-language data. It uses discrete token representation of VQGAN to facilitate image/video generation tasks. It demonstrates competitive performance in long-context retrieval tasks. It also reports comprehensive experimental results in both image/video understanding and generation tasks.

**Strengths:**

+ This paper is the first to scale sequence models up to 1 million tokens, pushing the boundaries of long-context processing in both language and multimodal (text-video) tasks.
+ The model use VQGAN tokenization to enable image/video understanding as well as generation.
+ In multi-needle retrieval tasks with a 128K context, the model achieves results comparable to or better than GPT-4, demonstrating its effectiveness in long-context retrieval tasks.

**Weaknesses:**

+ Lack of hour-long video benchmark evaluation, please refer to questions.
+ No comparison with existing open-source LLMs in context length of 128K, for example Llama 3.1
+ The usage of VQGAN decrease the image understanding ability as Table 4 shows.
+ There is no quantitative result on image generation tasks.

**Questions:**

1. The three benchmarks used to report video performance is not long. It is better to show accuracy on long-video benchmarks such as VideoMME[1] and MLVU[2].
2. In Table 3, how about comparing with existing open-sourced LLM supporting context length of 128K?
3. The paper is the first to scale context length up to 1M. However, there is no baseline comparison in 1M context length. How about using linear scaling technique during inference stage to construct a naive baseline? E.g., scale 8x from a 128k model.

[1] Video-MME: The First-Ever Comprehensive Evaluation Benchmark of Multi-modal LLMs in Video Analysis. arXiv preprint arXiv:2405.21075.
[2] MLVU: A Comprehensive Benchmark for Multi-Task Long Video Understanding. arXiv preprint arXiv:2406.04264.

---

> ### Author Response · Authors · 2024-11-18
> **Rebuttal**
>
> We thank the reviewer for helpful feedback. We provide detailed answers to the questions below.
>
> ---
> > **Long-video benchmarks**
>
> The reviewer requested an evaluation on long-video benchmarks. We evaluated LWM-1M on the Video-MME benchmark, and the results are shown in the table below. LWM-1M achieves the **highest performance in its model size category**, outperforming similarly sized models and approaching the performance of models ten times larger, and catching up with proprietary models like GPT-4o and Gemini-1.5-Pro.
>
> | **Model**        | **Parameters** | **Frames**       | **Medium (4min~15min)** | **Long (30min~60min)** |
> |-------------------|----------------|-------------------|--------------------------|-------------------------|
> | Gemini 1.5 Pro    | Unknown        | up to 1800        | 74.3                    | 67.4                   |
> | GPT-4o            | Unknown        | 384               | 70.3                    | 65.3                   |
> | LLaVA-Video       | 72B            | 64                | 68.9                    | 61.5                   |
> | VideoLLaMA 2      | 72B            | 32                | 59.9                    | 57.6                   |
> | Long-LLaVA        | 7B             | 64                | 51.4                    | 45.4                   |
> | Video-LLaVA       | 7B             | 8                 | 38.1                    | 36.2                   |
> | **LWM-1M**        | 7B             | up to 1800        | **63.7**                | **60.8**               |
>
>
>
> ---
>
> > **Comparing with existing open-sourced LLM supporting context length of 128K in needle retrieval tasks**
>
> Firstly, this work was completed some time ago, prior to the availability of existing open-source LLMs with a context length of 128K, such as Llama 3 (proof can be given once we may fully de-anonymize our work). We believe our work has inspired subsequent research on scaling context size.
>
> Secondly, at the time of this work, all open-source and proprietary LLMs suffered from the “lost in the middle”  issue [1], where models including GPT-3.5, Claude, and Llama performed poorly on needle retrieval tasks. Our naively trained model exhibited the same issue; hence, we proposed a data synthesization technique, which significantly improved our model to achieve near-perfect accuracy on these tasks.
>
> [1] Liu, N. F., Lin, K., Hewitt, J., Paranjape, A., Bevilacqua, M., Petroni, F., & Liang, P. (2024). Lost in the middle: How language models use long contexts. Transactions of the Association for Computational Linguistics, 12, 157-173.

---

> > ### Comment · Reviewer_cWVz · 2024-11-24
> >
> > Thanks for your response.
> >
> > It is totally understood that the work could in the case that completed earlier and  inspired a lot of future work. However, it is encouraged to compare with open-sourced long context LLMs to demonstrate the effectiveness of ring attention. I would be of higher recommendation for it as a in-date paper being published in the conference next year. I sincerely appreciate the contribution this paper offers, but a more comprehensive comparison to in-date models would make this paper better. Hence, I will keep the current rating.
> >
> > Best,
> > Reviewer cWVz

---

> ### Author Response · Authors · 2024-11-26
> **Response to Reviewer cWVz**
>
> Dear Reviewer cWVz,
>
> Thank you for your positive feedback and for recognizing our paper's contributions.
>
> We agree with the reviewer’s point that comparing our work to recent state-of-the-arts can further strengthen the paper. Following this suggestion, we have updated the paper with in-date baseline comparisons against state-of-the-art open-source models for our multi-needle retrieval task and long video evaluations
>
> >**Multi-needle Retrieval**
>
> We have updated the paper with more results on multi-needle evaluations on recent SoTA open source models (LLaMA-3.1-8B-Instruct, Qwen2.5-7B-Instruct, and Mistral-7B-Instruct-v0.3). The results are shown in Table 2.  For convenience, we have copied the tables below. To extend these models to 1M context, we use linear positional embedding scaling (8x) for the LLaMA 3.1, and similarly extend the YaRN positional embeddings for Qwen and Mistral. Although these models perform well on lower contexts (32K / 128K), extrapolating to much longer contexts (1M) is much more challenging, seen by the large drop in retrieval performance for these models. We believe this further demonstrates the effectiveness of our training methodology with progressive training, model-based data synthetization, masked sequence packing and ringattention. **Our model significantly outperforms these recent state-of-the-art methods at 1M context, achieving 0.67-0.84 accuracy across different settings compared to 0.0-0.32 for baseline models.**
>
> | Model (32K Context)      | N = 2, R = 2 | N = 4, R = 1 | N = 4, R = 2 |
> |--------------------------|--------------|--------------|--------------|
> | Gemini Pro (02/23)       | 0.34         | 0.44         | 0.6          |
> | GPT-4-1106               | 0.97         | 0.95         | 0.9          |
> | Llama-3.1-8B-Instruct    | 0.87         | 0.95         | 0.93         |
> | Qwen2.5-7B-Instruct      | **1.0**      | **1.0**      | **0.97**     |
> | Mistral-7B-Instruct-v0.3 | 0.98         | 0.85         | 0.83         |
> | LWM-Text-1M (Ours)       | 0.84         | 0.97         | 0.84         |
>
> |    Model (128K Context)   | N = 2, R = 2 | N = 4, R = 1 | N = 4, R = 2 |
> |:------------------------:|:------------:|:------------:|:------------:|
> | Gemini Pro (02/23)       |      -       |      -       |       -      |
> | GPT-4-1106               |     0.92     |     0.80     |     0.82     |
> | Llama-3.1-8B-Instruct    |   **0.98**   |     0.91     |     0.87     |
> | Qwen2.5-7B-Instruct      |   **0.98**   |     0.80     |     0.90     |
> | Mistral-7B-Instruct-v0.3 |     0.85     |     0.75     |     0.68     |
> | LWM-Text-1M (Ours)       |     0.83     |   **0.98**   |     0.83     |
>
> |    Model (1M Context)    | N = 2, R = 2 | N = 4, R = 1 | N = 4, R = 2 |
> |:------------------------:|:------------:|:------------:|:------------:|
> | Gemini Pro (02/23)       |      -       |      -       |       -      |
> | GPT-4-1106               |       -      |       -      |       -      |
> | Llama-3.1-8B-Instruct    |     0.27     |     0.32     |     0.18     |
> | Qwen2.5-7B-Instruct      |      0.0     |      0.0     |      0.0     |
> | Mistral-7B-Instruct-v0.3 |     0.05     |     0.13     |     0.10     |
> | LWM-Text-1M (Ours)       |   **0.67**   |   **0.84**   |   **0.69**   |
>
> > **Long video evaluation**
>
> We evaluated these state-of-the-art models on long-video tasks. The long-video benchmark VideoMME remains challenging—even GPT-4o and Gemini 1.5 Pro achieve less than 70% accuracy. As the long-video results (paper’s table 4 and included in our previous response) show, **LWM-1M achieves the highest performance in its model size category, outperforming similarly sized state-of-the-art open-source models like LLaVA-Video and Long-LLaVA**. It approaches the performance of models ten times larger and achieves comparable results to leading proprietary models like GPT-4o and Gemini-1.5-Pro.
>
> We hope that these additional experiments address the reviewer’s concerns, and would be happy to further discuss any additional concerns.
>
> Best,
>
> Authors

---

### Official Review · Reviewer_zoBJ · 2024-11-02

**Soundness:** 4
**Presentation:** 4
**Contribution:** 2
**Rating:** 6
**Confidence:** 5

**Summary:**

This paper introduces a training process that gradually extends LLMs from a limited context length of 4K tokens to an extremely long context of millions of tokens. The two-stage training process — where the first stage focuses on long-context expansion and the second stage addresses image and video understanding and generation — enables the models to support multimodal understanding and generation. The empirically feasible training process, which integrates understanding and generation capabilities into one extensive model using VQ, is quite useful. The evaluations, which involves both linguistic and multimodal tasks, are comprehensive.

**Strengths:**

- The empirically feasible training process that integrates understanding and generation capabilities into one extensive model using VQ is quite useful, and the evaluations encompassing both linguistic and multimodal tasks are comprehensive.
- The settings and hyperparameters for training and evaluation are presented in a meticulous manner.

**Weaknesses:**

- One major contribution of this paper is extending the context length of existing LLMs to extremely long through progressive training, which is a valuable practical implementation. However, aside from the fundamental support of RingAttention, the technical improvements in this study are marginal.
- Building models that support both multimodal understanding and generation is interesting and meaningful. However, the simple incorporation of an off-the-shelf VQ model into the framework does not yield promising results. I believe that more in-depth analysis and improvements are needed in this direction.
- There is a lack of quantitative evaluation for multimodal lengthy content understanding. For example, can we evaluate understanding of image details by increasing resolutions with lengthy vision tokens? Additionally, there are long video understanding benchmarks that could be used to assess the proposed LWM-1M.
- Some crucial details in the main content of the paper, such as the evaluation data and the trainable parameters in each stage, are missing.
- Compared to existing and up-to-date VLMs (e.g., LLaVA-NeXT, VILA, Qwen-VL, CogVLM), the performance of LWM on image understanding and video understanding benchmarks is relatively low.

**Questions:**

- For the retrieval task in section 3.3.3, what is the exact evaluation benchmark used for the Multi Needle in a Haystack?
- What are the trainable parameters for each step in the stage 2?

---

> ### Author Response · Authors · 2024-11-18
> **Rebuttal (1/2)**
>
> We thank the reviewer for helpful feedback. We provide detailed answers to the questions below.
>
> ---
> > **Valuable contributions but marginal technical improvements**
>
> While we do provide comprehensive training details, these represent careful empirical investigations that advance our understanding of scaling language models to unprecedented context lengths. Similar to how the original GPT papers advanced the field through careful documentation of scaling strategies on Transformers, our work provides crucial methodological insights that enable training of truly long-context models using RingAttention.
>
> While RingAttention provides an important foundation by resolving memory constraints, our work goes significantly beyond this by solving the fundamental challenges of end-to-end training with million-token context windows. This represents a substantial leap forward in practical capability.
>
> This work makes several methodological contributions that collectively enable a new state-of-the-art in long-context modeling:
> * An open-source codebase and models that enable reproducibility and accelerate future research in this domain
> * Novel AI-based synthetic data generation techniques that demonstrably solve the previously observed "lost in the middle" problem
> * A masked packed sequence approach that eliminates cross-sequence attention artifacts, improving model robustness
> * A progressive training methodology that, for the first time, makes million-token context windows practically achievable
>
> Each of these contributions represents a significant technical advancement that was necessary to achieve our results. Together, they provide a comprehensive framework for training and deploying truly long-context language models, opening new possibilities for research and applications in this rapidly evolving field.
>
>
> ---
> > **Can we evaluate understanding of image details by increasing resolutions with lengthy vision tokens?**
>
> It's an interesting idea to use more tokens to increase resolutions. It may not be directly applicable to LWM, because LWM was trained with a fixed tokenizer that encodes at a resolution of 256. While one could theoretically split higher-resolution images into multiple 256-resolution segments, this approach might introduce artifacts at segment boundaries and wouldn't necessarily lead to deeper semantic understanding. However, this would be an interesting future research or application.
>
>
> ---
> > **Long-video benchmarks**
>
> The reviewer requested an evaluation on long-video benchmarks. We evaluated LWM-1M on the Video-MME benchmark, which tests video understanding across different duration ranges. Our results demonstrate LWM-1M's strong performance despite its efficient 7B parameter size. Specifically, LWM-1M achieves scores of **63.7% and 60.8% on medium and long videos respectively, outperforming other 7B models by substantial margins (e.g., +12.3% over Long-LLaVA on medium-length videos)**. While larger 72B models show stronger performance, LWM-1M's results are notable given its 10x smaller parameter count. The model also shows competitive performance compared to recent proprietary models, though we note evaluation conditions may differ. The results show that LWM-1M achieves the **highest performance in its model size category**, outperforming similarly sized models and approaching the performance of models ten times larger, and catching up with proprietary models like GPT-4o and Gemini-1.5-Pro.
>
>
> | **Model**        | **Parameters** | **Frames**       | **Medium (4min~15min)** | **Long (30min~60min)** |
> |-------------------|----------------|-------------------|--------------------------|-------------------------|
> | Gemini 1.5 Pro    | Unknown        | up to 1800        | 74.3                    | 67.4                   |
> | GPT-4o            | Unknown        | 384               | 70.3                    | 65.3                   |
> | LLaVA-Video       | 72B            | 64                | 68.9                    | 61.5                   |
> | VideoLLaMA 2      | 72B            | 32                | 59.9                    | 57.6                   |
> | Long-LLaVA        | 7B             | 64                | 51.4                    | 45.4                   |
> | Video-LLaVA       | 7B             | 8                 | 38.1                    | 36.2                   |
> | **LWM-1M**        | 7B             | up to 1800        | **63.7**                | **60.8**               |

---

> ### Author Response · Authors · 2024-11-18
> **Rebuttal (2/2)**
>
> ---
> > **Details on Multi Needle in a Haystack**
>
> We built on top of the original Needle in a Haystack problem (see Section 3.3.2 for details), which was to retrieve a single magic number randomly inserted into some long context. Further details on the multi-needle retrieval evaluation was added to Appendix D, and also shown below for convenience.
>
> In the case of single need evaluation, magic numbers are associated with random cities (“The magic number for San Francisco is 2521233”  -> “What is the magic number for San Francisco?”). We extend the task by introducing two variables N and R, where N is the number of needles (magic numbers + different cities) randomly inserted into the context, and R is the random subset of magic numbers asked to retrieve (“What are the magic numbers for San Francisco, …, and Bangkok?”). Correctness is computed by extracting out the numbers retrieved for each city and checked with string matching.
>
> ---
> > **Other evaluation data and trainable parameters**
>
> For text and image understanding tasks, and short video analysis, we follow the data and evaluation process used in each respective paper. For short and long YouTube video QA, detailed evaluation protocols are described in Appendix G. For needle retrieval tasks, comprehensive evaluation details are provided in Section 3.3.2 and Appendix D.
>
> During training, all parameters are trainable for all stages.
>
>
> ---
> > **Results of visual understanding are not SOTA compared with up-to-date VLMs**
>
> While we acknowledge that our visual understanding results do not match current SOTA VLMs, this is not the primary focus of our work. Our contribution centers on pioneering end-to-end solutions for scaling context sizes up to 1 million tokens on long video and book sequences - a significant advancement that has not been achieved before. The relatively lower performance in visual understanding can be attributed to our deliberate architectural choices: we opted for a simpler backbone model and an older tokenizer to focus on solving the fundamental scaling challenges. This tradeoff allowed us to tackle the more complex problem of extreme sequence length processing. Using better data and more training are all that’s needed to achieve SoTA performance, as shown in the Chameleon paper. Future work can certainly incorporate more advanced visual encoders, more training, and better data to improve performance while building on our scaling innovations.

---

### Official Review · Reviewer_QidU · 2024-11-03

**Soundness:** 2
**Presentation:** 2
**Contribution:** 2
**Rating:** 3
**Confidence:** 5

**Summary:**

This paper introduces the Large World Model (LWM), a LLaMA-based model capable of processing sequences of up to 1 million tokens, combining language and video. The paper details the training datasets, progressive training stages, and evaluation results. It incorporates techniques such as RingAttention, Blockwise Transformers, and others for efficient training on million-length multimodal sequences.

**Strengths:**

1. The paper provides extensive training details on scaling token sequences to 1M tokens, pushing the boundaries of context length for multimodal models.
2. Empirical results demonstrate strong performance on challenging tasks like long video understanding and retrieval across 1M token contexts.
3. The paper is well-structured and clearly written, with detailed explanations of the model architecture, training process, and evaluation results.

**Weaknesses:**

1. The title "Large World Model" is misleading. The paper primarily explores training multi-modal (text and video) large language models with long context windows, without adequately defining or discussing the concept of a "large world model" or its relationship to MLLMs.
2. The paper fails to address fundamental questions about the necessity of long-context multi-modal LLMs for large world models and whether such models are sufficient for this purpose.
3. The paper largely describes training datasets and stages, lacking in-depth ablation studies and analysis. It relies heavily on previously proposed components like blockwise RingAttention, making it more akin to an experimental record than a research paper.
4. There is insufficient analysis of the potential synergies or conflicts between text and video in joint training. Do text and video in joint training mutually benefit each other, or do they have a negative influence on one another?
5. The paper lacks comprehensive comparisons with recent multimodal LLM works.
6. Despite claiming contributions in areas like masked sequence packing and loss weighting, the paper lacks ablation studies on those components

**Questions:**

The core issue is the unclear relationship between the concept of a large world model and the multi-modal LLM with long context window presented in the paper. Additional suggestions include:
1. Provide quantitative evaluations for image and video generation, as mentioned in Section 4.3.3.
2. Explain the larger fluctuations in the purple curve compared to other curves in Fig. 9.
To improve the paper, the authors should:
● Clearly define and discuss the concept of a "large world model" and its relationship to the proposed multi-modal LLM.
● Conduct and present ablation studies to demonstrate the impact of individual components.
● Provide more in-depth analysis of the interaction between text and video modalities during training.
● Include comprehensive comparisons with other recent multimodal LLM works.

---

> ### Author Response · Authors · 2024-11-18
> **Rebuttal (1/2)**
>
> We thank the reviewer for helpful feedback. We provide detailed answers to the questions below.
>
> ---
> > **Defining or discussing the concept of a "large world model"**
>
> Thank you for raising this important point about terminology. We use the term 'Large World Model' to describe our approach for several specific reasons:
>
> 1. The term 'World Model' has precedent in the field, starting with Ha & Schmidhuber's seminal work [1] which established that world models should capture environment dynamics and enable prediction. Our model similarly aims to capture and predict both linguistic and visual dynamics of the world through both language and video generation.
> 2. The qualifier 'Large' specifically refers to our model's extended context window, which enables modeling significantly longer sequences than previous approaches. This capability is crucial because real-world phenomena often unfold over long time horizons with complex dependencies.
> 3. While we acknowledge that our model does not capture all aspects of the world (and we make no claims of completeness), it advances world modeling capabilities by:
>     * Jointly modeling language and video dynamics
>     * Handling extended temporal dependencies
>     * Enabling predictive modeling over longer horizons
>
> This usage aligns with recent developments in the field, such as OpenAI's Sora being described as a "World Simulator" [2], while being precise about our specific contribution to this line of research.
>
> [1] Ha, David, and Jürgen Schmidhuber. "World Models." NeurIPS 2018
>
> [2] Brooks, Peebles, et al. "Video Generation Models as World Simulators." 2024
>
>
> ---
> > **Lacks ablation studies on masked sequence packing and data balancing**
>
> We thank the reviewer for raising this point about ablation studies. We apologize if this was not clear in our initial presentation, but we have indeed conducted comprehensive ablation studies on masked sequence packing, with the full results presented in Appendix B. Specifically, Table 8 shows the performance comparison with and without masked sequence packing, demonstrating an **10%~30% improvement** in multiple QA tasks. Table 10 shows the comparison with and without proper mixing of synthetic and chat data, showing a trade-off between synthetic data and chat data. To address this concern, we have now added explicit references to these ablation studies in Section 4.2 of the main paper and clarified their key findings.
>
>
> ---
> > **The paper lacks comprehensive comparisons with recent multimodal LLM works.**
>
> The reviewer is concerned that this work does not compare with recent multimodal LLM advancements. This work was completed some time ago, and since then, more multimodal long-context LLMs have been developed. Once the anonymous review period concludes, we can provide evidence that this work predates more recent multimodal long-context models, such as Gemini 1.5 Pro.
>
> Nonetheless, we appreciate the reviewer's suggestion and conducted more experiments to compare with recent works. We evaluated LWM-1M on the Video-MME benchmark, which tests video understanding across different duration ranges. Our results demonstrate LWM-1M's strong performance despite its efficient 7B parameter size. Specifically, LWM-1M achieves scores of 63.7% and 60.8% on medium and long videos respectively, outperforming other 7B models by substantial margins (e.g., **+12.3% over Long-LLaVA** on medium-length videos). While larger 72B models show stronger performance, LWM-1M's results are notable given its 10x smaller parameter count. The model also shows competitive performance compared to recent proprietary models, though we note evaluation conditions may differ. The results show that LWM-1M achieves the **highest performance in its model size category**, outperforming similarly sized models and approaching the performance of models ten times larger, and catching up with proprietary models like GPT-4o and Gemini-1.5-Pro.
>
> | **Model**        | **Parameters** | **Frames**       | **Medium (4min~15min)** | **Long (30min~60min)** |
> |-------------------|----------------|-------------------|--------------------------|-------------------------|
> | Gemini 1.5 Pro    | Unknown        | up to 1800        | 74.3                    | 67.4                   |
> | GPT-4o            | Unknown        | 384               | 70.3                    | 65.3                   |
> | LLaVA-Video       | 72B            | 64                | 68.9                    | 61.5                   |
> | VideoLLaMA 2      | 72B            | 32                | 59.9                    | 57.6                   |
> | Long-LLaVA        | 7B             | 64                | 51.4                    | 45.4                   |
> | Video-LLaVA       | 7B             | 8                 | 38.1                    | 36.2                   |
> | **LWM-1M**        | 7B             | up to 1800        | **63.7**                | **60.8**               |

---

> ### Author Response · Authors · 2024-11-18
> **Rebuttal (2/2)**
>
> ---
> > **The paper largely describes training datasets and stages, lacking in-depth ablation studies and analysis. It relies heavily on previously proposed components like blockwise RingAttention, making it more akin to an experimental record than a research paper.**
>
> The reviewer questions the contributions of this paper, characterizing it as 'more akin to an experimental record than a research paper.' We respectfully disagree with this assessment.
>
> While we do provide comprehensive training details, these represent careful empirical investigations that advance our understanding of scaling language models to unprecedented context lengths. Similar to how the original GPT papers advanced the field through careful documentation of scaling strategies on Transformers, our work provides crucial methodological insights that enable training of truly long-context models using RingAttention.
>
> While RingAttention provides an important foundation by resolving memory constraints, our work goes significantly beyond this by solving the fundamental challenges of end-to-end training with million-token context windows. This represents a substantial leap forward in practical capability.
>
> This work makes several methodological contributions that collectively enable a new state-of-the-art in long-context modeling:
> * An open-source codebase and models that enable reproducibility and accelerate future research in this domain
> * Novel model-based synthetic data generation techniques that demonstrably solve the "lost in the middle" attention problem
> * A masked packed sequence approach that eliminates cross-sequence attention artifacts, improving model robustness
> * A progressive training methodology that, for the first time, makes million-token context windows practically achievable
>
> Each of these contributions represents a significant technical advancement that was necessary to achieve our results. Together, they provide a comprehensive framework for training and deploying truly long-context language models, opening new possibilities for research and applications in this rapidly evolving field.
>
> ---
> > **There is insufficient analysis of the potential synergies or conflicts between text and video in joint training. Do text and video in joint training mutually benefit each other, or do they have a negative influence on one another?**
>
> We respectfully disagree with the reviewer's assessment. Our work directly addresses the challenges of text-video interactions in joint training through multiple technical innovations. Specifically:
> 1. We empirically observed that naive combination indeed leads to negative interference, which motivated our systematic approach to this problem.
> 2. The masked packed sequence technique we propose is explicitly designed to manage these interactions and reduce negative transfer.
> 3. Our ablation studies demonstrate that these mechanisms improve performance on both text and video tasks compared to baseline joint training approaches.
>
> While we agree that understanding the fundamental nature of text-video synergies remains an important open research question, our work makes concrete technical progress by establishing effective methods for joint training that mitigate negative interference. This creates a foundation for future research into positive transfer mechanisms.

---

> ### Author Response · Authors · 2024-12-01
> **Followup on Rebuttal**
>
> Dear Reviewer QidU,
>
> Thank you for taking the time to review our paper. We believe our rebuttal has addressed the reviewer’s concerns including terminology, ablation studies, and comparative evaluation. We would greatly appreciate your consideration of our response.
>
> Best,
>
> Authors

---

### Official Review · Reviewer_gZJr · 2024-11-04

**Soundness:** 3
**Presentation:** 3
**Contribution:** 4
**Rating:** 8
**Confidence:** 4

**Summary:**

This paper proposes a novel model architecture and training process that achieves significant advancements in long-context modeling for developing large language and vision-language models capable of processing sequences up to 1 million tokens in length. The authors propose a two-stage training process - in the first stage, they progressively train a base language model on increasingly longer text sequences, while in the second stage, the models incorporate vision capabilities through joint training on image and video data. The consequent Large World Model(LWM) family demonstrates impressive performance on a few challenging tasks such as long context retrieval and long video understanding.

**Strengths:**

1. The paper addresses an important technical challenge in building foundational AI models, which is the long sequence understanding.
2. This work achieves multiple state-of-the-art results across different training stages
3. The open-sourced implementation and pre-trained models benefit and accelerate progress in the research community.

**Weaknesses:**

The model's suboptimal performance on image and short video understanding tasks and the choice of visual encoding constrain the general applicability of this work.

**Questions:**

1. When training the models of LWM-1K and LWM-8K, there was an 16% mix of the pure text data added from OpenLLaMA in the batch, how was it determined and how sensitive is the model's performance to this parameter?
2. How did you achieve loss balancing across different tasks and contexts? Any specific strategy?

---

> ### Author Response · Authors · 2024-11-18
> **Rebuttal**
>
> We thank the reviewer for helpful feedback. We provide detailed answers to the questions below.
>
> ---
> > **How was OpenLlaMA data mix rate determined**
>
>
> Our data mix was primarily determined through iterative experimentation of training a little bit on different mixes (50% text, 25% text, 16% text, 8% text) and evaluating performance on standard language eval benchmarks (e.g. short text in Table 1). In the end, we decided on 16% as a proportion that would still enable faster training on more vision data per batch without a large sacrifice in text accuracy, which was important to retain for later on video-chat finetuning.
>
> ---
> > **Specific strategy about loss balancing across tasks and contexts**
>
> We performed loss balancing by separating each language or vision-language task into their own batch elements, as each task could have a very different proportion of vision / language tokens. For example, if random language data (100% tokens are text losses) was packed in the same batch element as a long-video language task (<1% tokens are text losses), averaging together to compute the text loss would give the long-video text loss very little weight. Thus, vision / text tokens losses would first be averaged within each batch element, and then averaged over the batch dimension.
>
> In addition, we add loss reweighting within each batch element to address some loss imbalance that would occur between the same task examples. For example, image-text QA questions might have a single text token output (True / False), while others would have many text tokens (detailed image description). Loss reweighting was done by applying a multiplier that would give each example equal weight, meaning higher per-token weight on shorter sequences, and lower per-token weight on longer ones. This was quite important, especially when using our model for some downstream image QA tasks (e.g. POPE) that had single token True/False outputs.

---

### Author Response · Authors · 2024-12-03
**Discussion Period Summary by Authors**

We thank the reviewers for taking the time to review our paper and considering our rebuttals. Here, we provide a summary of the reviews, rebuttals, and responses over the discussion period.

&nbsp;

Reviewers gave positive evaluations on our paper, saying that it:
* Addresses important technical challenges in learning general purpose foundation models by being the **first to scale to 1M context** **(gZJr, QiDU, cWVz)**
* Provides a public codebase, as well as detailed experimental and training details that will **greatly benefit the open-source community** **(gZJr, QiDU, zoBJ)**
* Provides a **simple, unified model design** that supports both understanding and generation over language, image, and video **(zoBJ, cWVz)**
* Demonstrates **state-of-the-art** results in **long-context** tasks **(gZJr, QiDU, cWVz)**

&nbsp;

During the rebuttal period, we addressed reviewers’ concerns with further discussion and experimental results:

* **Long VideoMME benchmark (QiDU, zoBJ, cWVz)**: LWM-1M achieves scores of 63.7% and 60.8% on medium and long videos respectively, __outperforming other similar sized models by substantial margins__ (e.g., +12.3% over Long-LLaVA on medium-length videos). These results establish it as **the highest performance in the 7B model category**, approaching the capabilities of 70B+ models and competitive with proprietary solutions like GPT-4V and Gemini-1.5-Pro.
* **Multi-Needle evals on state-of-the-art models (cWVz)**: Although baseline models perform well at lower contexts, extrapolating to 1M context proves difficult, showing on average 0% - 32% accuracy. In contrast, our model **significantly outperforms these recent state-of-the-art methods at 1M context**, achieving 67% - 84% accuracy across different settings.

&nbsp;

Best Regards,

Authors

---

### Meta-Review · Area_Chair_JAZg · 2024-12-21

**Metareview:**

### Summary:
This paper introduces LWM (Large World Model), a novel approach for processing million-length sequences in both language and video modalities. The key contribution is demonstrating scalable training of models that can handle extremely long context windows (up to 1M tokens) while maintaining competitive performance, particularly in video understanding tasks.

### Strengths:
1. Novel technical solution for scaling context length
> "The paper addresses an important technical challenge in building foundational AI models, which is the long sequence understanding." - gZJr

2. Comprehensive empirical validation
> "Empirical results demonstrate strong performance on challenging tasks like long video understanding and retrieval across 1M token contexts." - gZJr

3. Open-source contribution with practical impact
> "The open-sourced implementation and pre-trained models benefit and accelerate progress in the research community." - gZJr

### Weaknesses:
1. Limited image/video understanding capabilities
> "The model's suboptimal performance on image and short video understanding tasks and the choice of visual encoding constrain the general applicability of this work." - gZJr

2. Insufficient comparative analysis
> "The paper lacks comprehensive comparisons with recent multimodal LLM works." - cWVz

3. Questions about fundamental necessity
> "The paper fails to address fundamental questions about the necessity of long-context multi-modal LLMs for large world models and whether such models are sufficient for this purpose." - QidU

### Justification:

Despite the noted weaknesses, I recommend acceptance because:
1. The paper makes significant technical contributions in scaling context length - a fundamental challenge in AI systems - with demonstrated improvements over existing approaches
2. The comprehensive empirical validation and open-source implementation provide substantial value to the research community
3. The authors have thoroughly addressed most reviewer concerns during the discussion period with additional experiments and clarifications
4. While image/video understanding performance could be improved, this limitation is acknowledged and does not diminish the core contribution of enabling million-token context processing

The reviewers are have carefully evaluated the technical details. The authors have provided thorough responses with additional experimental validation.

**Additional Comments On Reviewer Discussion:**

- Reviewer QidU did not engage with authors' detailed rebuttal addressing terminology, ablation studies, and comparative evaluations. Therefore less weight should be given to these concerns.
- Authors provided substantial new experimental results comparing with recent models like LLaMA-3 and comprehensive ablation studies showing 10-30% improvements from their techniques.
- Authors convincingly demonstrated strong performance on long-video tasks, with LWM-1M achieving the highest performance in its model size category.

---

### Decision · Program_Chairs · 2025-01-22

Accept (Poster)